# Multi-annual patterns of rapidly draining supraglacial lakes in Northeast Greenland

Katrina Lutz[1], Ilaria Tabone[1, 2], Angelika Humbert[3, 4], Matthias Braun[1]

[1]Institute of Geography, Friedrich-Alexander-Universität Erlangen-Nürnberg, Erlangen, 91058 Germany
[2]Department of Geophysics, University of Concepción, Concepción, Chile
[3]Alfred Wegener Institute Helmholtz Centre for Polar and Marine Research, Bremerhaven, 27570 Germany
[4]Department of Geosciences, University of Bremen, Bremen, 28359 Germany

*Correspondence to*: Katrina Lutz (katrina.lutz@fau.de)

**Abstract.** Supraglacial lakes are known to undergo rapid drainages in which their water masses are drained through ice hydrofracture to the glacier bed, typically within several hours. Despite the impact of this sudden englacial transport of meltwater, the conditions leading up to a rapid drainage are not fully understood. In this study, the spatial and temporal variability of rapid drainages was evaluated over two major glaciers in Northeast Greenland: Zachariæ Isstrøm and Nioghalvfjerdsfjorden (79N Glacier). Over the 2016 - 2022 summer melt seasons, supraglacial lakes on these glaciers were tracked via Sentinel-2 optical imagery to find the occurrence of any rapid drainages. The spatial distribution of rapid drainages as well as the seasonal timings were then evaluated against several other factors, such as ice strain rate, elevation, lake volume and seasonal surface temperature. It was found that the drainage patterns of individual lakes varied substantially, with some lakes having drained only a couple times and others nearly every year in the observed time frame. Furthermore, the temporal and spatial consistency of drainages were also generally inconsistent. Some lakes tended to drain at similar times over the melt years, while others had a more sporadic drainage timing. Similarly, certain clusters of lakes tend to drain in similar time frames, whereas it was found that most lakes did not follow a drainage tendency based on spatial position. However, the phenomenon of cascading drainages, in which neighboring lakes drain nearly simultaneously, was observed several times. While it was seen that drainages tend to occur later with higher elevations, little correlation was found between the occurrence of rapid drainages and the other investigated factors. It appears several conditions would need to be met to allow for a rapid drainage to occur, particularly the existence of fractures or crevasses within the lake boundaries.

## 1 Introduction

The role of supraglacial lakes (SGLs) on glacier mechanics, subglacial dynamics and ocean chemistry has become increasingly more intriguing in recent years, as the interconnectedness of supra- and subglacial hydrology has become more understood (Andrews et al., 2018; Khan et al., 2024; Koziol and Arnold, 2018). Supraglacial lakes develop seasonally in glacial surface depressions and store surface meltwater before either draining or refreezing. Through sudden hydrofracture events, SGLs can undergo rapid drainages, where a lake partially or fully drains within hours to days (Das et al., 2008). The meltwater contained

in these lakes is then routed towards the bed through moulins or crevasses to the glacier bedrock or is stored englacially. Meltwater flow to the bed ceases if the channel freezes or is closed due to surrounding pressure. It has been observed that this sudden influx of water to the bedrock can cause local and temporary glacier uplift and increased ice velocity due to decreased basal friction (Andrews et al., 2018; Das et al., 2008; Dow et al., 2015; Doyle et al., 2013; Joughin et al., 2013; Neckel et al.,
2020; Stevens et al., 2022; Tedesco et al., 2013). Furthermore, a rapid drainage can increase local tensile stresses due to increased basal pressure from the drained water flowing along inefficient subglacial drainage pathways (Stevens et al., 2015). This increased tensile stress from one drainage can trigger cascading drainages in the surrounding area due to subsequent fractures (Christoffersen et al., 2018; Chudley et al., 2019; Otto et al., 2022), triggering widespread flow acceleration (Maier et al., 2023). Inland expansion of surface meltwater has increased by roughly 30% throughout Greenland over the last few
decades (Tedstone and Machguth, 2022), and is expected to persist as surface air temperatures and thus surface melt area are predicted to continue increasing in the coming decades (Hanna et al., 2021). Currently, around a third% of SGLs in western Greenland undergo rapid drainage (Chudley et al., 2019; Cooley and Christoffersen, 2017; Fitzpatrick et al., 2013), , producing significant perturbations in subglacial water inflow that will become only more frequent and widespread.

To better understand the causes and effects of rapid drainages, various research groups have conducted in situ observations of rapid drainages using seismometers, GNSS sensors, water-level sensors and optical imagery, mostly in southwest Greenland (Chudley et al., 2019; Das et al., 2008; Dow et al., 2015; Doyle et al., 2013; Stevens et al., 2015; Tedesco et al., 2013), but also recently in the southeast (Stevens et al., 2022) and in the northeast sector of the ice sheet (Neckel et al., 2020). Through these observations, the dynamics of a rapid drainage can be seen, including the seismic activity caused by ice fracture, glacial
uplift, ice velocity changes and high water discharge rates. While detailed information about individual drainages is beneficial for understanding localized conditions surrounding and leading up to them, large-scale observations allow for the identification of patterns over space and time. Several large-scale, remote sensing analyses were recently conducted over different glaciers in Greenland for one melt season (Miles et al., 2017; Williamson et al., 2018) and for extended time periods (Fitzpatrick et al., 2013; Liang et al., 2012; Morriss et al., 2013; Otto et al., 2022). In these studies, the timing and location of
rapid drainages are identified. Through the three time series analyses, warmer surface temperatures were seen to shift drainage timing slightly earlier and increase lake altitude due to the more extensive presence of surface meltwater at higher elevations with warmer surface temperatures; however, minimal further correlation among drainage timing and location was found.

Despite these previous large-scale rapid drainage studies primarily conducted in southwest Greenland, there is still a lack of
understanding of the potential influencing factors on rapid drainages, especially in the understudied region of northeast Greenland. This region is especially intriguing as it is expected to show the most significant inland expansion of SGLs within Greenland in the coming decades (Ignéczi et al., 2016), which could have profound impacts on regional ice flow and freshwater influx into the ocean. This research aims to address this knowledge gap by analyzing the spatial and temporal distribution of rapid drainages in northeast Greenland over the 2016 to 2022 summer melt seasons. By utilizing high resolution (10 m)

Sentinel-2 optical data, we will track lake development, quantify lake volume trends, and evaluate environmental and mechanical factors influencing rapid drainage occurrence and timing. This evaluation will provide valuable insight to help improve our understanding of the role of SGLs in this region and their potential impacts on glacier dynamics and hydrology.

## 2 Study area

In this study, we focus our analysis on Zachariæ Isstrøm and Nioghalvfjerdsfjorden (79N Glacier) in Northeast Greenland

(Fig. 1a). These two large, fast-flowing glaciers drain roughly 12% of the Greenland Ice Sheet (Mouginot et al., 2015; Rignot and Mouginot, 2012), which contains a 1.1 m sea level rise equivalent (Morlighem et al., 2014). These marine-terminating glaciers extend roughly 600 km into the ice sheet (Khan et al., 2022) and diverge around the Lambert Land nunatak at the coast. 79N Glacier has a floating tongue of over 80 km in length and 20 km in width, whereas Zachariæ Isstrøm's floating tongue disintegrated between 2002 and 2015, leading to a drastic increase in rate of retreat (Mouginot et al., 2015). These areas

are also known for high levels of surface melt, visible through the roughly 900 supraglacial lakes developing there annually Lutz et al. (2024). We study the region containing the ablation zone upstream of both glaciers' grounding lines, similar to Hochreuther et al. (2021) and Lutz et al. (2024), where Zachariæ Isstrøm's grounding line is roughly along its calving front.

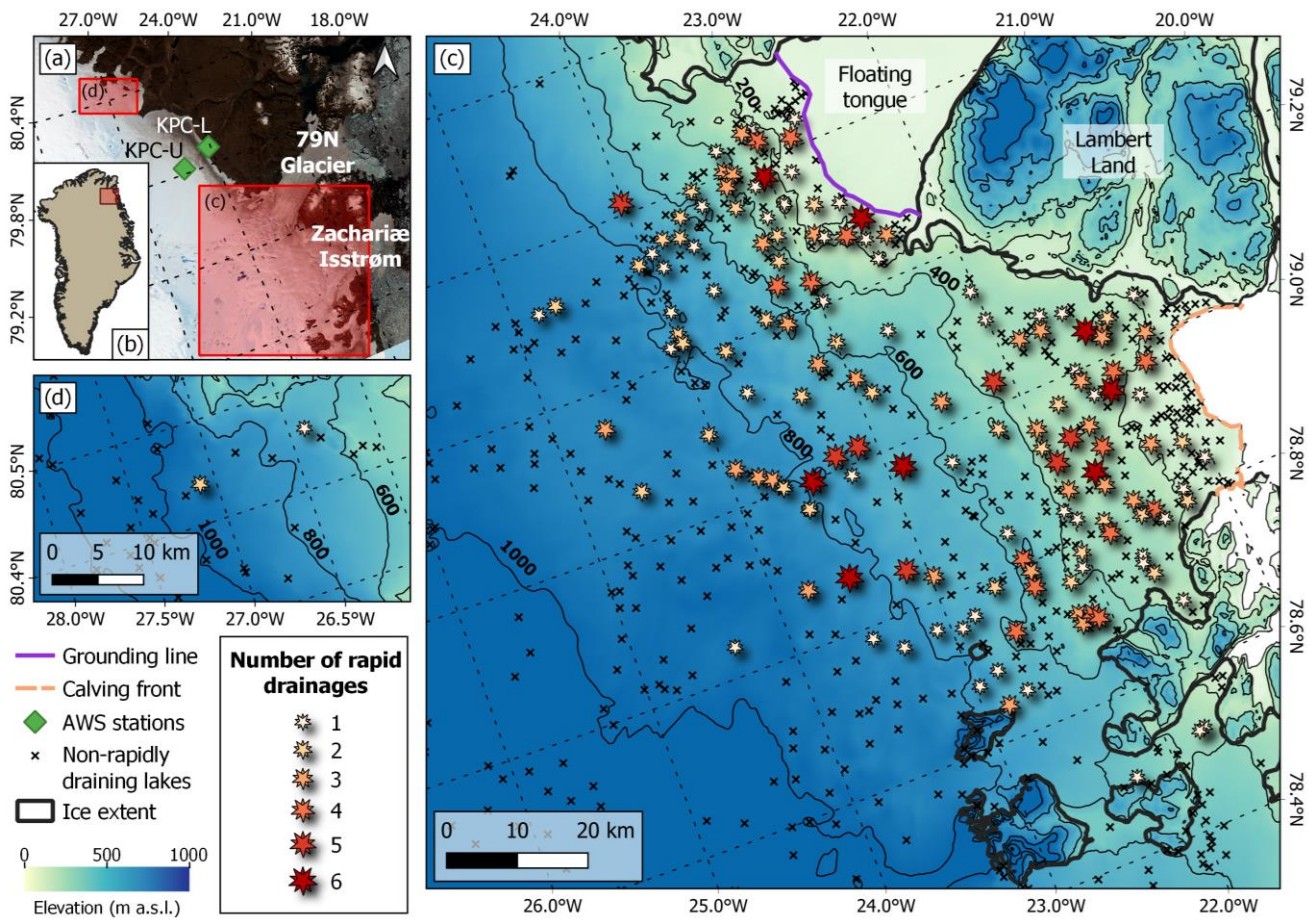

**Figure 1: (a) An overview of the study area in Northeast Greenland showing a Sentinel-2 image from 3 August 2020. (b) An overview of the study region in the context of Greenland. (c) and (d) are subsets of (a) showing the spatial distribution of lakes in the region. Lakes marked with an 'x' have not rapidly drained and those marked with a star show the number of times they have drained over the 2016 - 2022 summer melt seasons. These subsets are underlaid with ArcticDEM elevation data (Porter et al., 2022).**

## 3 Data and methods

To enable the lakes to be easily tracked over several melt seasons, each lake in our area of interest was assigned an identification number. Tracking specific lakes is possible since the surface depressions where lakes form are directly influenced by bedrock topography (Gudmundsson, 2003; Lampkin and Vanderberg, 2011), and thus stay in the same location. In this study, we tracked the development of the lakes at a near daily rate over the 2016 to 2022 summer melt seasons (roughly June to September) using the visible bands from Sentinel-2 L2A imagery, as developed in Lutz et al. (2023). With this time series, individual lakes were semi-automatically reviewed for sudden drops in area and were subsequently manually checked using Sentinel-2 imagery to verify the occurrence of a rapid drainage. Due to a lack of daily cloud-free imagery, it was often difficult to determine the exact date of the drainage, so the drainage date was set as the day on which the lake first appeared to be

drained in available imagery. In consistency with previous work (Das et al., 2008; Doyle et al., 2013; Tedesco et al., 2013; Williamson et al., 2018), we define the timeframe of a rapid drainage as observed through remote sensing methods to be two days or less. However, during periods lacking cloud-free imagery, this threshold was extended to encompass the entire data-free period, with a maximum gap of ten days. Almost all drainages, however, were determined based on short image intervals, with an average interval of 2.8 days among all rapid drainages. In order to accommodate partial rapid drainages (Chudley et al., 2019), we implemented a low volume loss threshold of 50% and manually checked to remove any false positives.

To assess the amount of water lost through each rapid drainage, a reflectance-depth algorithm developed in Lutz et al. (2024) was used, where a depth value is calculated for each lake pixel in the green band of a Sentinel-2 L2A image. This equation is described as

$$z = 14.9572e^{-4.2629x} + 0.5242 \,, \tag{1}$$

where $z$ is depth and $x$ is green reflectance. Through this, the volumes of the lake pre-drainage and post-drainage are calculated by multiplying by the pixel size of 10 m by 10 m. The difference of these volumes is the amount of water lost through rapid drainage, to be either transported to the glacier bed or stored englacially.

Furthermore, to investigate potential influencing factors for rapid drainages, we acquired ice surface velocity, elevation and temperature data for the region. The ice surface velocity was downloaded from the National Ice and Snow Data Center's MEaSUREs Greenland Monthly Ice Sheet Velocity Mosaics from SAR and Landsat program (https://nsidc.org/data/nsidc-0731/versions/5). In this study, the velocity data used were calculated over the month of July 2019 using Landsat mosaics for the purpose of calculating strain rate in a high melt year. Each lake was assigned the ice surface velocity at its center; however, if velocity data directly over the lake was unavailable, the nearest available velocity point was used. From this velocity data, the principal strain rates of the ice, $\epsilon_{1/2}$, were calculated as

$$\epsilon_{1/2} = \frac{\epsilon_{xx}+\epsilon_{yy}}{2} \pm \sqrt{\left(\frac{\epsilon_{xx}+\epsilon_{yy}}{2}\right)^2 + \left(\frac{\epsilon_{xy}}{2}\right)^2} \,, \tag{2}$$

where $\epsilon_{1/2}$ is the first and second principal strain rates, $\epsilon_{xx}$ $\epsilon_{yy}$ and $\epsilon_{xy}$ are components of the strain rate tensor, and $\epsilon_{xy}$ is known as the shear strain rate.

Additionally, the 100 m resolution data from ArcticDEM (https://www.pgc.umn.edu/data/arcticdem/) was similarly used to assign an elevation to the center of each lake. Finally, the temperature data was taken from two automatic weather stations (AWSs) that are part of the PROMICE program: KPC-L and KPC-U (How et al., 2022). The locations of these weather stations are roughly 100 km northwest of the main area of interest, as shown in Fig. 1a. In this study, the monthly surface temperatures for June, July, and August are averaged for each year to provide a yearly summer average for each station.

# 4 Results

## 4.1 Spatial and temporal distribution of rapid drainages

The spatial distribution of rapidly draining lakes can be seen in Fig. 1, where the size and color of the marker indicates how many years out of seven each specific lake drained. Additionally, lakes that filled at least one year but did not drain between 2016 and 2022 are marked with an 'x'. In total, 864 lakes within our area of interest filled up at least once in the seven year period, 152 of which rapidly drained at least once. Most drainages tend to be found at lower elevations; only two rapidly draining lakes are found above 900 m. When only considering elevations below 900 m, 32.1% of lakes have undergone rapid

drainage.

The temporal and spatial variations found for each lake and across the region are shown in Fig. 2 and detailed in Fig. 3. Here, each lake is represented by a wheel, within which each segment is colored to represent the week in which the lake drained for each year. Additionally, if the lake never filled up in a certain year, it is marked with an 'x' to highlight the fact that the lake

did not drain that year because there was nothing to drain in contrast to a lake not draining due to other factors. In Fig. 2, a few general tendencies can be seen, albeit there are exceptions to these statements. Firstly, lakes tend to drain a bit later in the season at higher elevations and earlier at lower elevations, particularly on Zachariæ Isstrøm. Excluding 2018, drainages above 400 m occur on average 10.66 days later than drainages below 400 m in the same melt season (see Fig. A1). This pattern is supported by the fact that lakes at higher elevations only tend to develop later in the melt season, thus limiting the possibility

of drainages to after having filled, as supported by previous studies (Doyle et al., 2013; Fitzpatrick et al., 2013). Additionally, there was only one drainage recorded above elevations of 400 m in 2018, which was a particularly cold and dry year, compared to the 45 drainages that occurred above 400 m in 2019. Further, out of the 26 drainages in 2018, only six of those lakes had also drained in 2017, implying that the majority of the drainages in 2018 had accumulated water from both melt seasons before draining. Another remark can be made about lakes that often do not fill up for one or more years after a drainage. Out of the

seven investigated years, a few small clusters of lakes follow this pattern by not filling with meltwater for several years, examples of which are highlighted in Fig. 3f and g. Since these lakes are dispersed across slow and fast moving areas, this implies some other underlying mechanism than ice deformation keeping the drainage channels in these areas open.

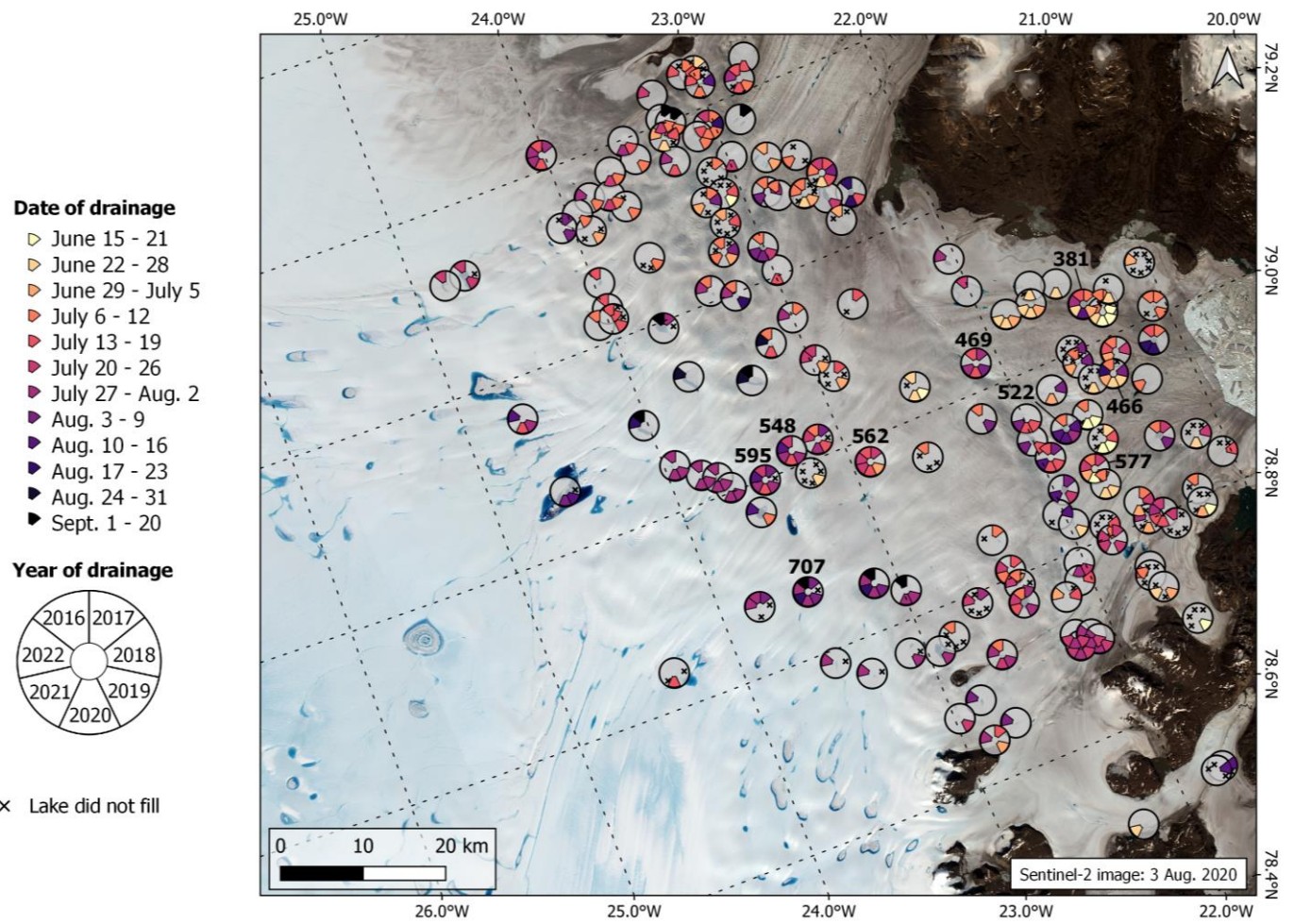

**Figure 2: The spatial and temporal distribution of rapid drainages in Northeast Greenland. Each rapidly draining lake is identified with a wheel, where each segment represents one of the seven years over the 2016 - 2022 summer melt seasons. Each segment is filled with a color corresponding to the week in which it drained that specific year. If a segment is empty, the lake filled but did not drain. If a segment is marked with an 'x', the lake never filled that year. Frequently draining lakes that are used in later analyses are labeled with their ID numbers.**

Another aspect to be noted is the temporal variability of drainages at each lake. While many lakes tend to drain within a relatively consistent time span (roughly within the same three weeks), others have a seemingly random drainage pattern (see Fig. A2). Furthermore, there are clusters of lakes that follow similar drainage patterns and timings. Figure 3b, d and e show groups of lakes that all tend to drain within the same few days in years when they do drain. For example, the group of lakes in Fig. 3d tends to drain quite early in the year, roughly within the last week of June, while the lakes in Fig. 3e drain near the end of July.

Additionally, it can be seen that neighboring lakes will occasionally drain near=simultaneously in cascading events. Fig. 3c shows an extensive cascading drainage event in which eight lakes within roughly 17 km of each other drained on the same day

in 2019, along with a couple of lakes draining within days of this event. Several lakes in Fig. 3b also drain simultaneously, where cascading drainages are seen over a few different years. However, not all lakes drain simultaneously each time. For example, the three lakes on the left (lakes 565, 584, and 587) drain together in 2016, but the lake directly to the right of them (lake 598) only drains during the cascading drainages in 2019 and 2020. Additionally, lakes 595 and 548drain several more times than the other lakes, without triggering drainages in them, perhaps due to their position downstream of the others.

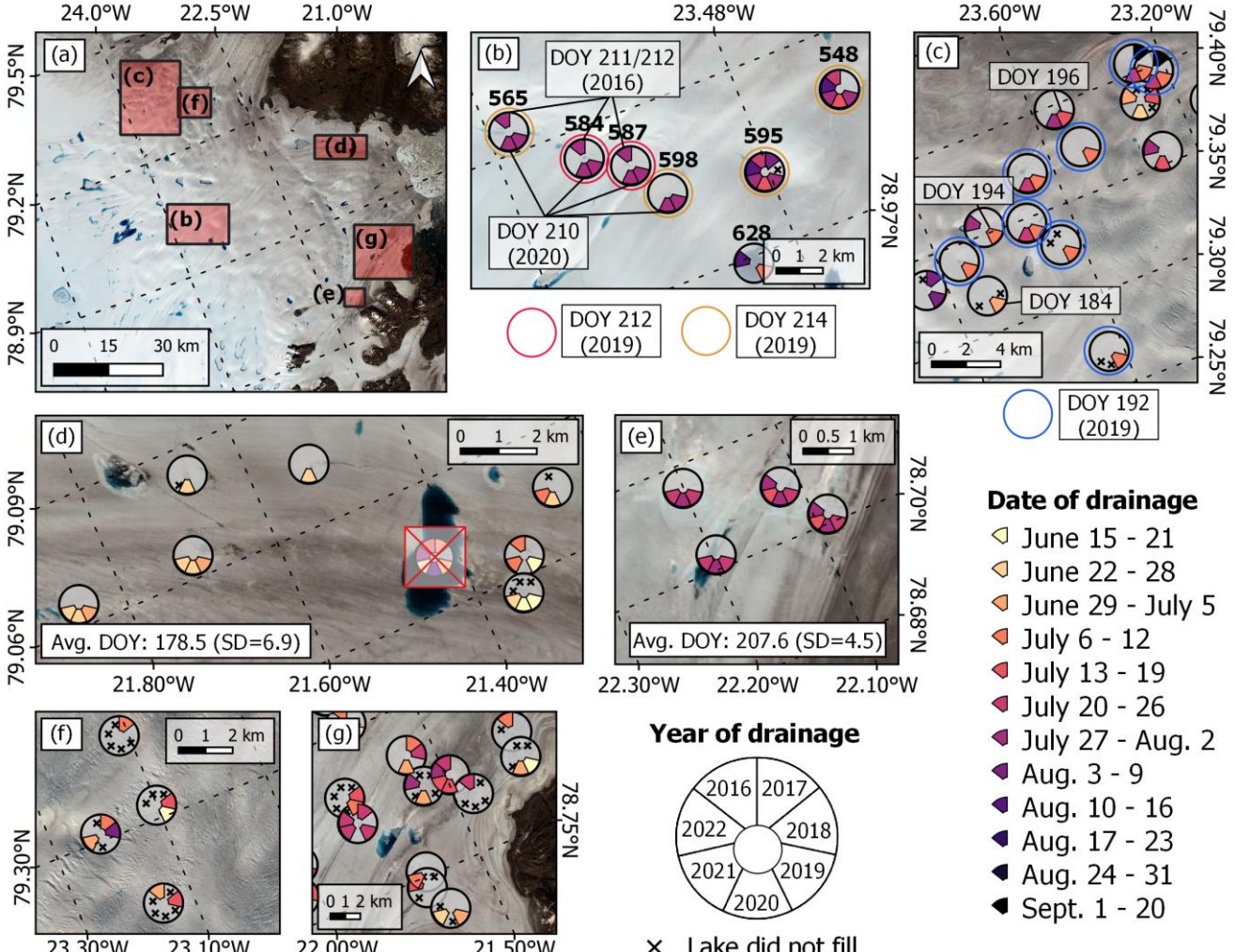

**Figure 3: Detailed subsets from Fig. 2. (a)** An overview of the subsets from the area of interest. **(b)** A group of lakes that undergo cascading drainages in three different years: 2016, 2019, and 2020. The three lakes that drained simultaneously in 2016 are marked with arrows and the day of year (DOY). The four lakes that drained simultaneously in 2020 are similarly marked. The six lakes that drained nearly simultaneously in 2019 are circled with either red (for DOY 212) or orange (for DOY 214). **(c)** A cascading drainage of several lakes on DOY 192 in 2019, marked by blue circles. Three lakes that drained within a few days of this cascading drainage are individually marked on the map. **(d)** and **(e)** Groups of lakes that exhibit similar drainage timing. The average DOY and standard

deviation (SD) for all drainages in each subset, except the lake crossed out and not included because of its distinctly different drainage pattern, is shown. (f) and (g) Groups of lakes that often do not refill, the years for which are marked by 'x'. The background for all subsets is a Sentinel-2 image from 3 August 2020.

## 4.2 Potential factors contributing to rapid drainages

Several factors were investigated to determine if one or more environmental conditions or lake characteristics would trigger a rapid drainage after a certain threshold is passed or under other certain conditions. A hypothesis used in models(Alley et al., 2005; Arnold et al., 2014; Banwell et al., 2013; Banwell et al., 2016; Koziol et al., 2017; Tsai and Rice, 2010; van der Veen, 2007) but also recently contradicted by observations (Fitzpatrick et al., 2013; Stevens et al., 2015; Williamson et al., 2018) is the presumption that each lake has a specific volume limit, above which the pressure from the water causes a hydrofracture-induced drainage. In Fig. 4, we show the results of nine often-draining lakes that were investigated to determine the volume of water in the lake preceding a rapid drainage and their maximum volume in seasons when they did not drain. The location of these lakes on the glaciers can be found in Fig. 2. It can be seen that oftentimes the lakes drained at relatively low volumes, while having reached larger volumes in years where it did not drain, exemplarily seen with Lake 577. There are, however, a couple of lakes where the lowest volume occurred in a year in which the lake did not drain (e.g. Lake 562). These low volumes occurred in 2018, which was a remarkably low melt year. This pattern, however, is not substantiated since Lake 548 also had low volumes that year but also did not drain on another year in which it reached its maximum volume. Furthermore, the volumes of the rapid drainages vary substantially throughout the years, with pre-drainage volumes, for example, for Lake 381 ranging from 0.0034 km$^3$ to 0.0269 km$^3$. These findings thus imply that each lake does not drain based solely on a volume threshold alone.

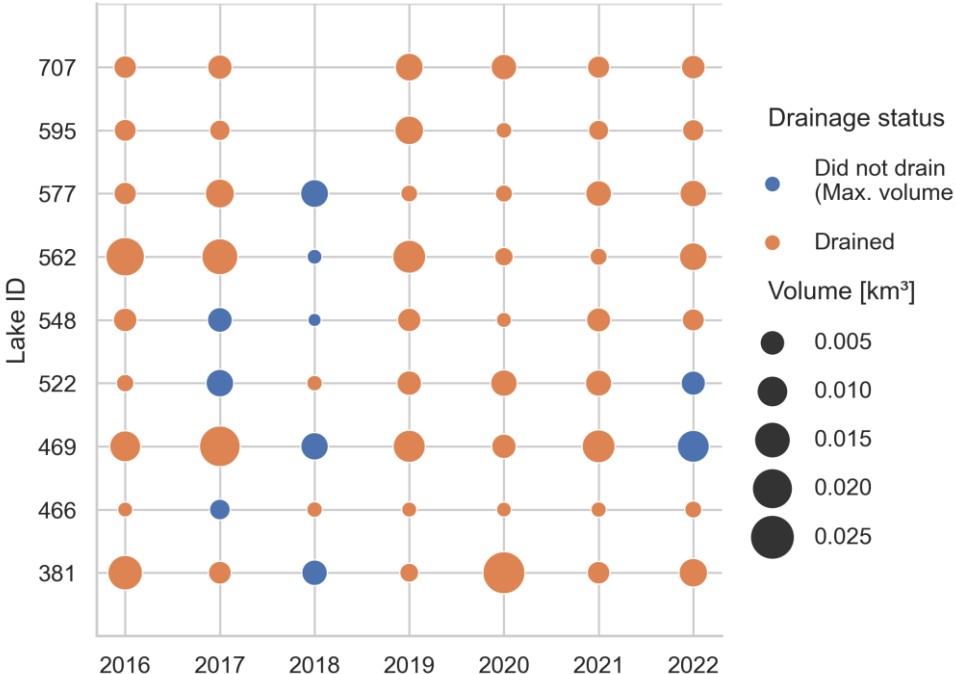

**Figure 4: The lake volume preceding rapid drainages for the nine most frequently draining lakes over the 2016 - 2022 summer melt seasons. The size of the circle indicates the amount of water volume present in the lake at the most recent observation before the rapid drainage (orange). For years where the lake did not rapidly drain, the maximum volume reached is recorded (blue). For lakes that did not fill up the entire melt season, no circle is present that year.**

Furthermore, the parameter of ice strain rate at the center of each lake was explored for potential influence on drainage frequency. The first and second principal ice strain rates are mapped in Fig. 5a and d, respectively, according to Eq. 2. The ice strain rates are generally low except for a few distinct regions. Figures 5b, c, e, and f show detailed panels of these high strain areas, where the location of all investigated lakes and how often they have rapidly drained are denoted. In Fig. 5c and f, two regions of high strain rate can be seen running parallel to the northern and southern edges of Zachariæ Isstrøm, where the first

principal strain is predominantly found around the northern edge of the glacier and the second principal strain is predominantly found around the southern edge. In these areas, there are several regularly rapidly draining lakes, but also many lakes that have not rapidly drained in the observed time frame. Similarly, there are two distinct lines of high strain parallel to the flow direction on 79N Glacier, along the eastern and western edges, highlighted in Fig. 5b and e. The first principal strain rate dominates along the eastern edge, where there is only one rapidly draining lake with several non-rapidly draining lakes. Large magnitudes

of both the first and second principal strain rates are found along the western edge, with the largest second principal strain rates concentrated in a line in the northern part of the subset. While there are several rapidly draining lakes in this region, there are even more non-rapidly draining lakes, and no distinct patterns are found in the areas of most concentrated strain rates. Generally, there does not appear to be a correlation between high strain rate and a high occurrence of rapid drainages.

This trend is further emphasized in Fig. 5g and h, where first and second principal ice strain rates and a drain-to-fill ratio are compared, respectively. This ratio was calculated for each lake, consisting of the number of drainages observed over the number of seasons the lake filled with water. This gives a more accurate sense of how often a lake drains given the opportunity, since there are often years where the drainage channel remains open, not allowing water to accumulate. To compare the strain data to temporally correlated drainages, the lakes that drained in the 2019 melt season are highlighted in red, since the strain

data is derived from velocity data from July 2019. Firstly, it can be seen that lakes that never rapidly drain (with a drain-to-fill ratio of zero) are found across ice with a wide range of strain rates. These lakes are found even in the areas where high strain rates are found. Furthermore, there is no distinct trend among the rapidly draining lakes in relation to strain rate. The majority of rapidly draining lakes are located in low strain rate regions, with an average first principal strain rate of 0.0047 $a^{-1}$ (SD = 0.0090 $a^{-1}$) and an average second principal strain rate of -0.0060 $a^{-1}$ (SD = 0.0092 $a^{-1}$). The non-rapidly draining lakes have a

similar strain rate distribution with an average of 0.0038 $a^{-1}$ (SD = 0.0079 $a^{-1}$) for the first principal strain rate and an average of -0.0039 $a^{-1}$ (SD = 0.0080 $a^{-1}$) for the second principal strain rate.

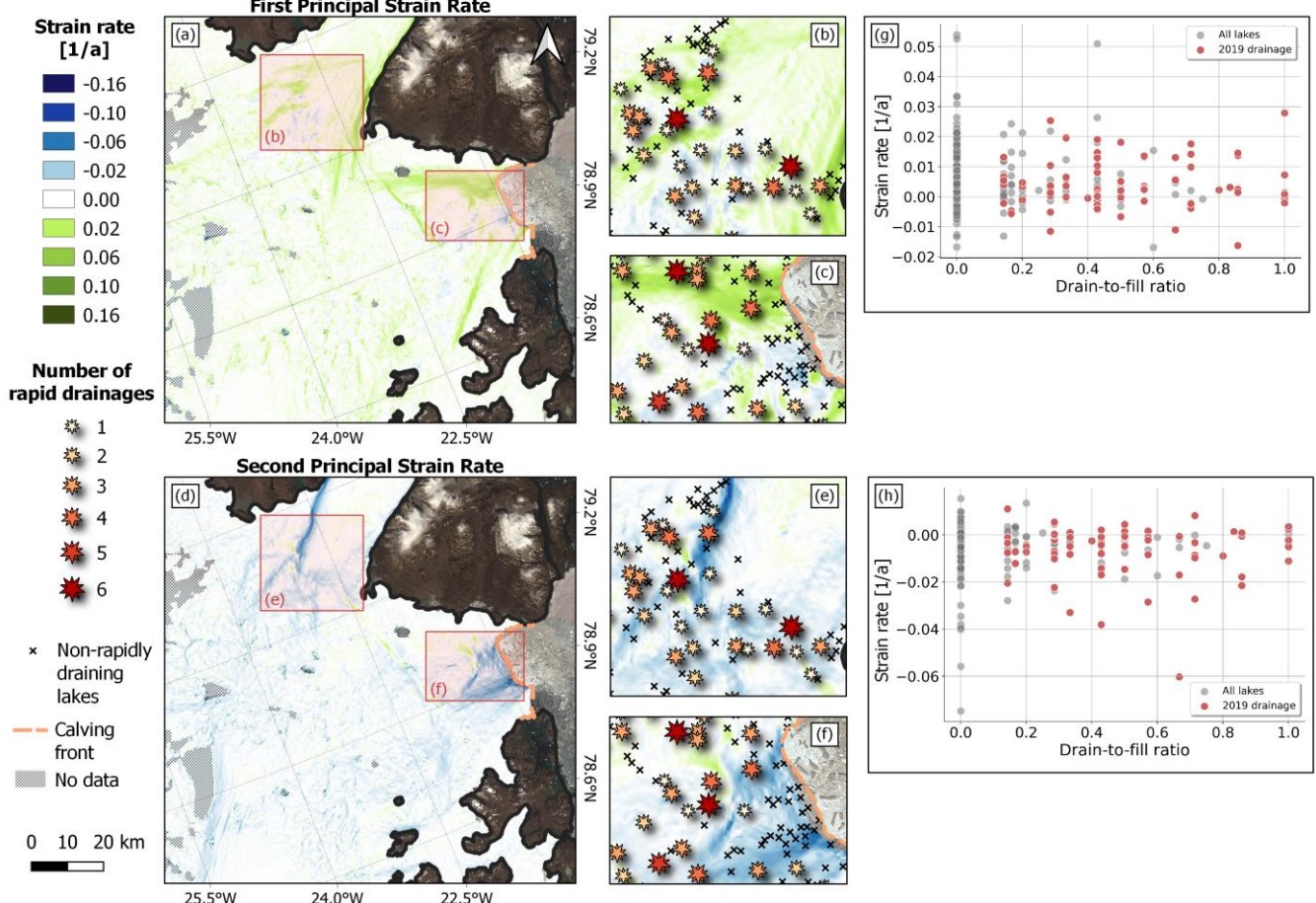

**Figure 5: (a) The first principal strain rate over the area of interest. (b) and (c) show subsets of (a) with the number of rapid drainages that each supraglacial lake underwent, with the lakes that have filled but not rapidly drained marked with an 'x'. (d), (e), and (f) mirror (a), (b), and (c) but show the second principal strain rate. (g) and (h) show the drain-to-fill ratio plotted against the first and second principal strain rates, respectively. This ratio is calculated as the number of times the lake has rapidly drained over the number of times the lake has filled over the 2016 - 2022 period, i.e. a lake that has drained only half of the times that it has filled has a ratio of 0.5. Lakes with a drain-to-fill ratio of zero never rapidly drain. The drainages that took place over the 2019 melt season are highlighted in red, since the strain data is derived from velocity data from July 2019.**

Finally, we investigated the impact of overall melt trends on drainage patterns. The average summer surface air temperature (June, July and August) for the AWS stations KPC-L and KPC-U (Fausto et al., 2021) is plotted in Fig. 6 for each year. These stations are located at 377 m a.s.l. (KPC-L) and 867 m a.s.l. (KPC-U). The mean local lapse rates were calculated between the stations for each melt season, allowing for the range of temperatures over elevations where lakes were present (44 to 942 m a.s.l.) to be estimated. Additionally, Fig. 6 shows the filling and drainage status of all lakes that have undergone a rapid drainage at least once in the 2016 - 2022 period, totaling to 152 lakes. These lakes are categorized by whether they underwent a rapid drainage, filled but did not rapidly drain, or did not fill at all that melt season. Here, a general tendency of more rapid drainage occurrences in warmer years can be seen; however, they are not always correlated. For example, the warm year of 2019 and the remarkably cold year of 2018 show the most and the fewest rapid drainages, respectively. However, despite being also relatively warm years, there were comparatively not as many rapid drainages in 2020 and 2021. In 2021, a larger number of lakes did not fill up at all, implying these drainage channels remained open throughout the melt season. This could contribute to the lower number of rapid drainages, but does not apply to the 2020 melt season, where roughly half as many lakes did not fill at all compared to 2021.

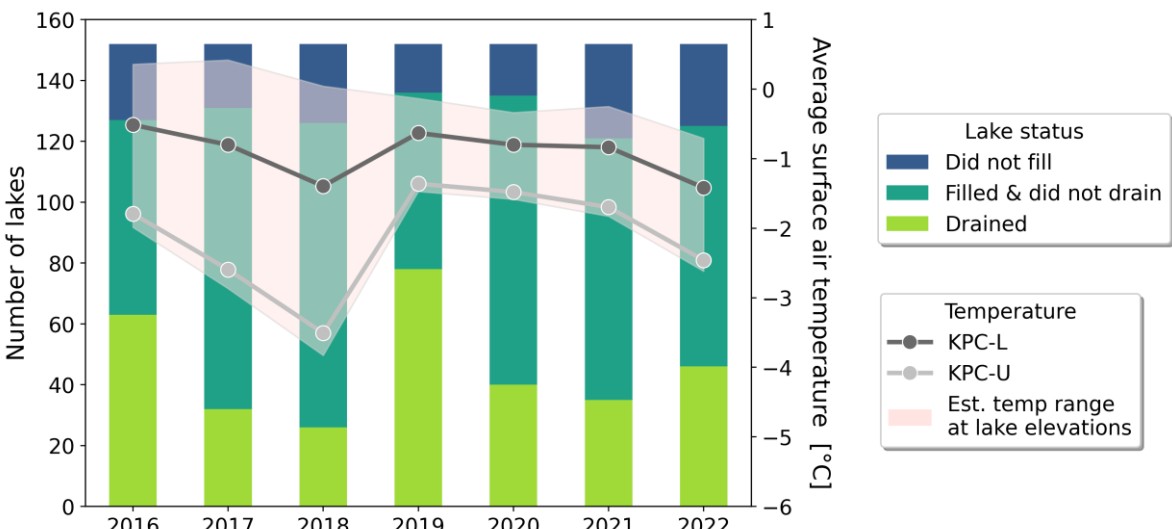

**Figure 6: On the left axis, the cumulative number of lakes that have drained at least once over the 2016 - 2022 period are categorically plotted in the bar graphs for each year, totaling 152 lakes. The lower category (light green) shows the number of lakes that underwent**

a rapid drainage that melt season. The middle category (green) shows the number of lakes that filled with water but did not rapidly drain that melt season. The upper category (blue) shows the number of lakes that did not fill up at all that melt season. On the right axis, the surface air temperature averaged over June, July and August for each year is plotted for both the KPC-L (dark gray line) and KPC-U (light gray line) weather stations. The shaded area shows the estimated temperature range for elevations of the analyzed lakes.

## 5 Discussion

Through this analysis, some patterns seen in other studies have been reaffirmed. Firstly, the correlation of drainage timing to elevation is apparent, where drainages tend to happen later at higher elevations (Doyle et al., 2013; Fitzpatrick et al., 2013; Miles et al., 2017). Here, it can be seen that the timing is dependent on the year and the specific melt development of the season, meaning general drainage timings cannot be made based on altitude alone, but the relative timing shift between lower elevation and higher elevation drainages can be seen within each year (see Fig. A1). Elevation alone, however, does not influence whether a lake will rapidly drain or not. In our study, both rapidly and non-rapidly draining lakes were found to be rather evenly distributed up to an elevation of 850 m, above which lakes rarely drained rapidly. Similar results were found by Williamson et al. (2018), where the correlation between elevation and rapidly or non-rapidly draining lakes was indistinguishable. Otto et al. (2022) support this finding but emphasize the expansion of rapid drainage occurrence to higher altitudes throughout their nearly four decade-long evaluation on Ryder Glacier. Furthermore, the investigation into ice strain rate resulted in no correlation between rapidly draining lakes and their location in regions of high or low ice strain rate for either the first or second principal strain rate. Williamson et al. (2018) also found no connection between whether a lake is rapidly draining or not and the principal strain rate or the von Mises yield criterion of the ice. In our study, there are also no clear patterns of rapidly draining lakes being located downstream of areas of high ice strain rate. Additionally, despite warmer melt seasons directly corresponding to greater lake extent (Lutz et al., 2023; Turton et al., 2021), warmer temperatures did not consistently cause an increase in rapid drainages, even though the most rapid drainages were seen in the warmest year (2019). This conclusion was also reached by Fitzpatrick et al. (2013) when conducting a decadal investigation of the Russell glacier catchment in western Greenland, despite also seeing the most rapid drainages in the peak melt year. In their study on the Sermeq Avangnardleq ice shed in western Greenland, Morriss et al. (2013) conclude that warmer temperatures cause more inconsistent rapid drainage behavior. One aspect to consider is that the environmental conditions of the previous year could reasonably have a substantial impact on the following year. For example, the particularly large amount of rapid drainages found in 2019 in our study could be the result of the subglacial pathways becoming less efficient in the previous cold year (2018) due to the decrease in meltwater flow. With such an increase in meltwater in 2019 into an inefficient subglacial drainage system, larger basal pressure from the drained meltwater could be inflicted, potentially initiating ice fracture and thus rapid drainages. This influence of a cold year on a subsequent warm year would explain the lower drainage occurrence in 2020 and 2021, where, despite also being relatively warm, the subglacial drainage system was still rather efficient due to the large outflow of meltwater in 2019.


Furthermore, the importance of the spatial proximity of SGLs in the context of rapid drainages has been highlighted in this study through the occurrence of so-called cascading drainages. Such cascading drainages events have been noted by other researchers (Christoffersen et al., 2018; Chudley et al., 2019; Fitzpatrick et al., 2013; Otto et al., 2022), who hypothesize that the tensile stresses in the surrounding ice increase due to the increase in basal water from one drainage, which then triggers

further drainages. Through remote sensing observations, it is difficult to assess the precise order of drainages in a cascading drainage, since the drainages can occur over the span of hours and there is generally only one satellite image available for the day. Thus, our study cannot confirm or deny the concept of trigger lakes and response lakes, as found in Chudley et al. (2019), but our findings do further emphasize the occurrence of many spatially close drainages within a short period of time. Additionally, through observing drainage patterns over several years, our study shows that the same combination of lakes in a

region are not necessarily involved in a cascading drainage every time one occurs (see Fig. 3b). This implies that either the ice stresses vary considerably year to year, and thus the increase in stress from one lake drainage does not necessarily create enough increased stress to cause other fractures to occur, or that there are other conditions needed for the neighboring lakes to participate in a cascading drainage. A potentially strong influence on the occurrence of cascading drainages is the efficiency of the subglacial drainage system and the resulting local tensile stresses. With larger basal pressure on the ice sheet, ice fracture

could occur more easily, which further amplifies basal slip and pressure, triggering hydrofracture in other nearby lakes as well. Such a phenomenon was observed in Stevens et al. (2015), where the increased injection of meltwater in the subglacial system from the drainage of a nearby lake caused local uplift hours before another hydrofracture-induced rapid drainage occurred. Furthermore, Otto et al. (2022) found that lakes located above large subglacial drainage paths tend to drain more frequently in cascading drainages, highlighting the influence of subglacial dynamics on drainage patterns.


One important factor that could cause irregularities in drainage patterns is the presence of crevasses within the lake boundary. As many areas of these glaciers have high ice velocities (see Fig. A3), a pre-existing crevasse could be quickly moved into and then out of a lake's boundaries, as seen in Chudley et al. (2019). For areas with lower ice strain rates, it could be difficult for a crevasse to form through ice fracture over the lake itself and the lake would thus only drain if a pre-existing crevasse

were to move into the lake while other conditions are fulfilled, e.g. high lake volume or increase in tensile stresses due to nearby drainage. Furthermore, for lakes in high strain areas, a crevasse formed in the center of the lake could theoretically be refractured the next melt season, given that the ice velocity is slow enough that the crevasse remains within the lake boundaries. Upon investigating this region, the half-width of many lakes in high strain regions is smaller than the distance the ice would move within a year, implying drainage by a refracture would be difficult. Many of the frequently draining lakes, however,

have a half-width large enough that a crevasse could remain within the lake boundaries the following melt season. As there are also other lakes with such large widths in the flow direction that do not drain frequently, a more detailed analysis of crevasse

formation and flow would be necessary. For this, the use of widespread very high resolution imagery over one or more melt seasons would be helpful in assessing the influence of crevasse presence in combination with other environmental factors.

**6 Conclusions and outlook**

In our study, rapid drainage patterns for supraglacial lakes in Northeast Greenland were investigated for the summer melt seasons of 2016 to 2022. While some trends were seen, many of the analyzed environmental and geographical parameters were found to be uncorrelated. Some clusters of lakes tend to have regular or consistent drainage patterns, but overall such lakes were in the minority. One noticeable trend is that rapid drainages tend to occur later at higher elevations, as these lakes only tend to fill up later in the season. However, no correlation was found between elevation and whether a lake rapidly drained or

not. Furthermore, no correlation was found between rapid drainages and ice strain rate, seasonal surface temperature, or a lake volume threshold. Despite no direct influence of individual parameters on rapid drainage occurrence being found, the observed phenomenon of cascading drainages implies a concrete trigger with influence over a larger spatial region. We would presume that a rapid drainage is triggered by the combination of several factors, most importantly including the presence of a crevasse within the lake boundary and large local basal stresses.


In order to further investigate the topic, higher resolution remote sensing data or multi-annual in situ data would be beneficial. With higher resolution remote sensing data, the presence of crevasses could more easily be tracked, allowing for a better assessment of the lack of drainages due to the absence of a drainage pathway. Additionally, with more detailed multi-annual in situ studies on several lakes, the influence of ice strain and other environmental conditions on rapid drainage could be better

understood. Overall, this study shows the variability of rapid drainage occurrence within individual lakes, among small lake clusters and across the entire region with no clear influence from the investigated factors individually.

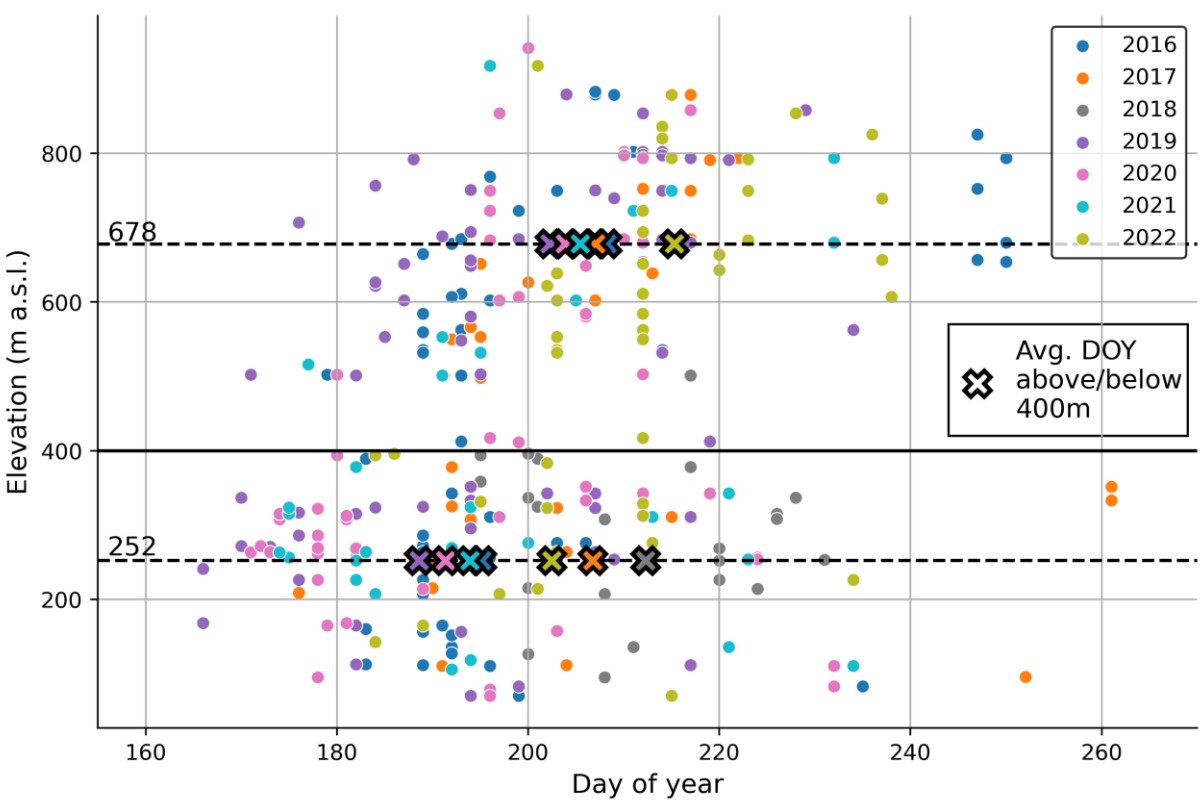

**Figure A1: The day of year (DOY) of each rapid drainage vs. the elevation of the center of the lake. These drainages are categorized**
**by color based on their year of drainage. The drainages are divided into elevation categories (above or below 400 m a.s.l.) by the**
**horizontal black line. For each year, the DOY of all lakes below and above 400 m are averaged separately and demarcated with an**
**'X', representing the average DOY for that year and elevation span. The average elevation for lakes below or above 400 m are**
**labeled and marked with a central horizontal dashed line. There is no average for 2018 above 400 m since only one drainage occurred**
**above that elevation that melt season.**

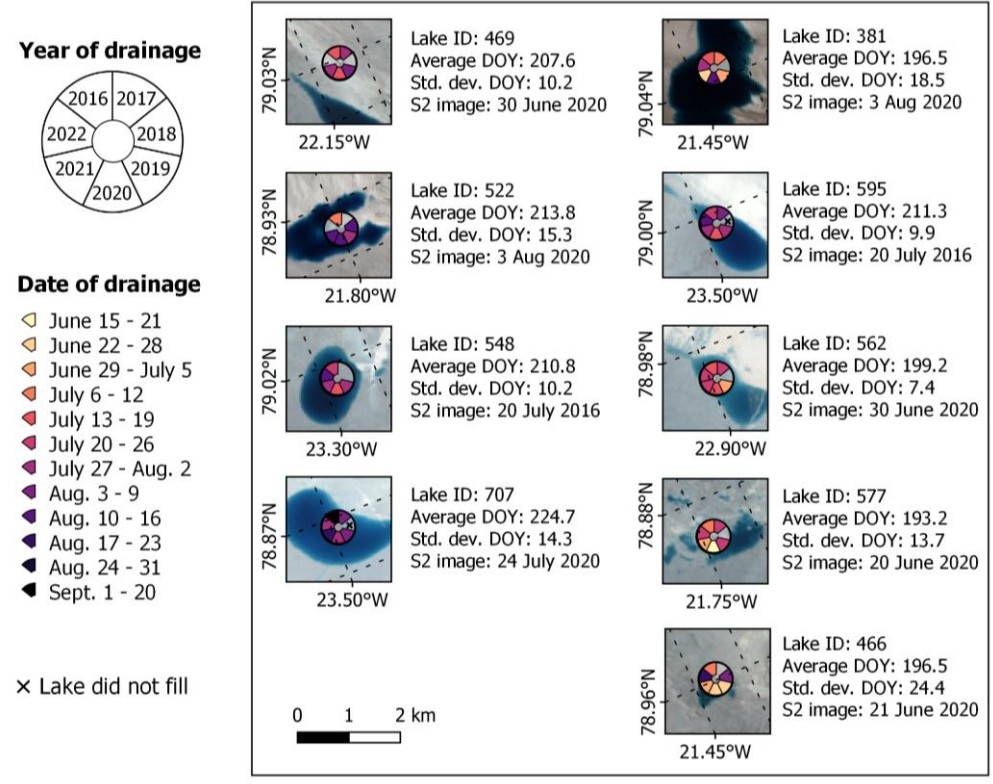


**Figure A2: Nine of the most frequently draining lakes in the area of interest. For each lake, the average day of the year (DOY) that it drained as well as the standard deviation among the days are noted. Years in which the lake did not fill with water are marked with an 'x'. The location of these lakes on the glaciers can be found in Fig. 2.**

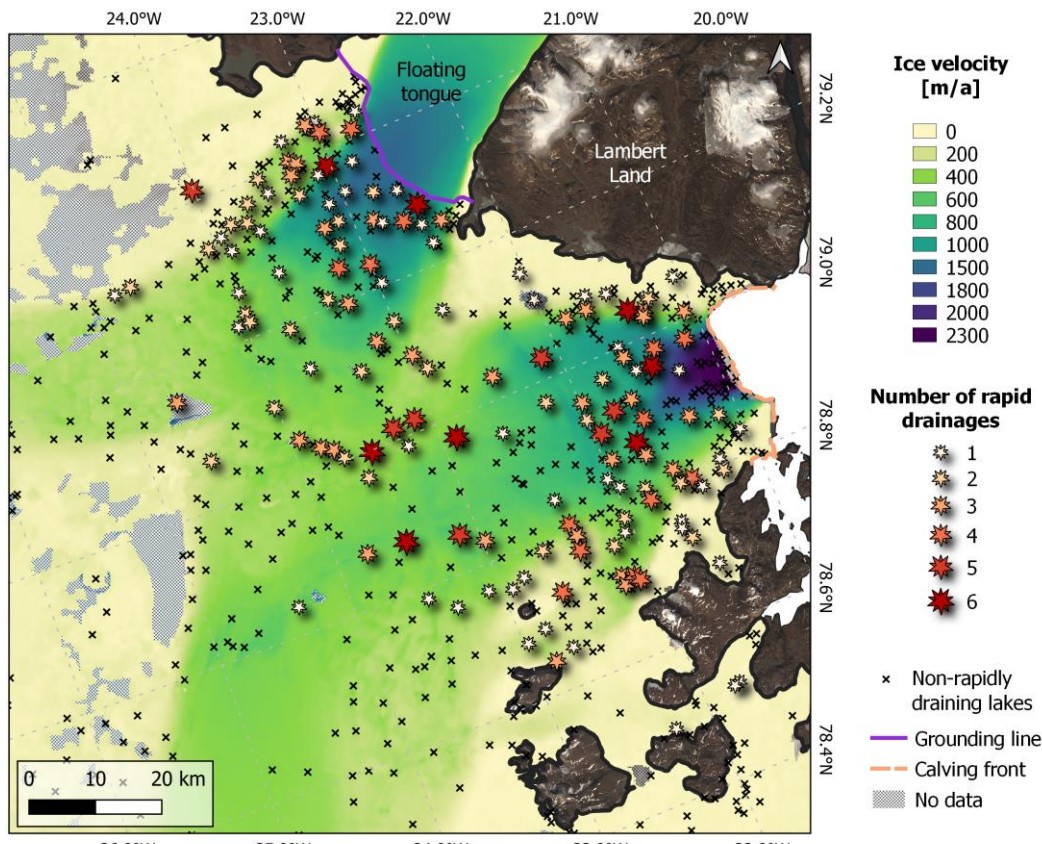

**Figure A3: The ice velocity magnitude data from MEaSURES from July 2019. Lakes marked with an 'x' have not rapidly drained and those marked with a star show the number of times they have drained over the 2016 - 2022 summer melt seasons.**

*Code and data availability:* The data relating to the location and timing of rapid drainages is freely available through the PANGAEA database under the following DOI: https://doi.org/10.1594/PANGAEA.973446.

*Author contributions:* KL and MB were responsible for conceptualization. KL curated the data, developed the methodology and conducted the formal analysis. KL and IT were responsible for the development of necessary code. All co-authors were involved in assessment of the results. KL was responsible for data visualization and preparation of the original draft. MB was responsible for funding acquisition, project administration, and supervision. All co-authors contributed to reviewing and editing the manuscript.

*Competing interests:* The authors declare that they have no conflict of interest.
*Acknowledgements:* We would like to thank ESA for providing Sentinel-2 data free of charge. The study was funded by the Bavarian State Ministry of Science and the Arts within the Elitenetwork Bavaria International Doctoral Programme "MOCCA - Measuring and Modelling Mountain Glaciers and Ice Caps in a Changing Climate". We also acknowledge financial support by the Deutsche Forschungsgemeinschaft and the Friedrich-Alexander-Universität Erlangen-Nürnberg within the funding

program "Open Access Publication Funding." Additionally, we would like to thank Luke Trusel for providing valuable insight into the interpretation of our results throughout the review process.

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
