# Peer review of "Multi-annual patterns of rapidly draining supraglacial lakes in Northeast Greenland"

_EGUsphere, 2024_

## Referee Comment (RC1)

**Multi-annual patterns of rapidly draining supraglacial lakes in Northeast Greenland**

**by Katrina Lutz, Ilaria Tabone, Angelika Humbert, Matthias Braun**

Comments by Nina Kirchner, Stockholm University and Tarfala Research Station, 27 December 2024

**Summary**

This manuscript presents mainly observational data in relation to drainage of supraglacial lakes on two outlet glaciers in north-eastern Greenland. The focus is on rapid, and furthermore, cascading (also known as coupled or, as here referred to: chained) drainage events during the years 2016-2022. Observed drainage patterns are based on analysis of Sentinel-2 L2A imagery. Timing of rapid drainage is reported to be related to elevation at which drainage takes place, confirming similar observations reported in the literature. It is further reported that investigations remain inconclusive regarding the relation between rapid drainage and i/lake volume prior to drainage, ii/ ice flow dynamics (via strain rate, represented by the July 2019 configuration for the 2016-2022 time-interval) around the time of rapid drainage, and iii/ averaged summer (JJA) surface air temperatures for each year for the collective drainages of each year.

**Assessment and specific comments:**

Detailed datasets on supraglacial lake drainage on glaciers in the northern sector of the Greenland Ice Sheet are few. Therefore, the here described novel dataset is of value and interest for the scientific community. A persistent challenge with observational datasets, especially from rarely studied regions, is that their perhaps largest value is their mere acquisition and open-access publication. This is often overlooked, and there is moreover a risk that works presenting and exploring novel datasets are rejected because they are regarded as lacking clear research questions and/or hypothesis. However, without base data, these are impossible to formulate.

To me, the manuscript by Lutz et al. is a telling example. A rich dataset is presented, and a first attempt is made to attribute rapid drainages to several potential "drivers", but no other conclusions can be established than there is great spatio-temporal variability, and likely not a single driver but multiple ones, acting possibly intertwined with exact mechanisms still to be discovered. Yet this is a solid conclusion. Yes, a statistical analysis of how potential drivers related to rapid drainage is not included in the analysis presented here, but since the dataset presented is – to me – the largest achievement, the author's "first approximation" analysis of attribution is sufficient in my view. As the dataset will be made freely available, it is for the scientific community to pick up on this when need is seen.

However, I think the manuscript would benefit from major revisions. To start with, a stronger motivation should be given why this dataset was created and analysed. Formulations such as "rapid drainages can substantially affect […] aspects of the glacial environment throughout the melt season" are too much of a commonplace and don't match the level expected from a manuscript submitted to TC. In the following, I recommend that a comment on terminology regarding the simultaneous (raid) drainage of supraglacial lakes is included. To my knowledge, these events have first been referred to as "cascading drainage" (Christoffersen et al., 2015, cited in the manuscript), then "coupled drainage (Otto et al., 2022. Supraglacial lake expansion, intensified lake drainage frequency, and first observation of coupled lake drainage, during 1985-2020 at Ryder Glacier, N Greenland. Front Earth

Sci, doi.org/10.3389/feart.2022.978137), and then here as "chain drainages". It would be beneficial in my view to adopt an already existing terminology (unless "chained" exists already, but in that case I miss a reference to it).

Continuing on this, it needs to be pointed out that the work of Otto et al. (2022), doing a similar study at Ryder Glacier, has apparently been overlooked. As Otto's work covers the time-span 1985-2020, including first observed coupled drainage at Ryder Glacier, I had expected to read about a comparison of drainage of SGLs at northern Greenland glaciers at least in the Discussion section?

Likewise, I am curious how rapid drainage events contrast to slow drainage events. I assume that to identify "rapid" drainage, all types of drainage need to be detected first? I realize that including a detailed comparison of slow vs rapid drainage is likely beyond the scope of the manuscript, but it would be helpful to know e.g how large the fraction of rapidly draining lakes is compared to slowly draining (or mentioning other similar simple comparative metrics). For southwest Greenland, such info is cited from the literature, but how is it for the region studied here? I think info is available (from e.g Fig 1 which indicates non-rapidly draining lakes), so could be included in a straightforward manner, and even be given some thought and reflection in the discussion section.

Finally, I think it is assumed in the manuscript (unless I misunderstood) that all drained water reaches the bed. But what about englacial storage, that is, if the drained volume vanishes from the surface but if not all of it reaches the bed? Could this impact subsequent drainage in one way or the other? If crevasses in the lake would aid rapid drainage, maybe englacial storage would hinder it? Maybe englacial storage could be reflected on in the discussion, I would be curious to hear the author's thoughts on it.

Concerning more editorial aspects, the title of the manuscript and the abstract reflects its contents. Overall, the presentation is well-structured and easy to follow, however, major work should be put in making the text more concise (detailed comments are given but may not be exhaustive). Care needs even to be taken concerning the use of time units throughout the manuscript, figures, and figure captions. The figures (main and supplementary) display a wealth of information (where more is desirable this is indicated in the detailed comments) and are still very appealing, great work has been done here!

**Detailed comments and questions:** Note that rewording suggestions are rendered in orange.

**Line 9-10:** "Supraglacial lakes are known to undergo rapid drainages in which the contents of the lake are drained through ice hydrofracture to the glacier bed"… -> "Supraglacial lakes are known to undergo rapid drainages in which their watermasses are drained through ice hydrofracture to the glacier bed…"

**Line 10-11:** "Despite the impact of this sudden loss of meltwater from the glacier, the conditions.. .." -> "Despite the impact of this sudden englacial transport of meltwater, the conditions…"

**Line 13:** "Over the 2016 - 2022 summer melt seasons, each supraglacial lake was tracked "--- -> "Over the 2016 - 2022 summer melt seasons, supraglacial lakes on these glaciers were tracked" Question: Over the entire glaciers or with focus on specific regions? This info comes later (l 63) but should be given here.

**Lines 17-20,** from "Furthermore": these sentences are unclear/vague and need to be reformulated

**Line 20:** physical location -> spatial position ?

**Line 20:** Chain drainage. I think you may be referring to "cascading" or "coupled drainage", as earlier described by Christoffersen et al., 205, and Otto et al., 2022. If yes, perhaps coupled drainage should be used here, too as terminology. If however chain drainage differs from coupled drainage, it should be explained (later) how.

**Line 20**: unclear wording in "in which more than one neighboring lake" – neighboring to what? You mena to another lake. You could rewrite as "in which neighboring lakes"

**Line 23:** "would need to be filled to allow for a rapid drainage to occur, particularly the existence of a crevasse within" -> "would need to be met to allow for a rapid drainage to occur, particularly the existence of fractures or crevasses within". In any case, I suggest that something in plural form is needed. Not just one crevasse.

**Line 27:** References are needed re better understood interconnectedness of supra- and subglacial hydrology

**Line 29:** "where the lakes" -> where a lake"

**Line 30:** "routed vertically" -> "routed towards the bed". Note that crevasses may not extend all the way to the bed, and that meltwater may be stored englacial. Note also that neither moulins nor crevasses are vertical.

**Line 31**: "Conduits remain open… until the channel freezes or is closed". I would rephrase this differently, like "meltwater supplies to bed ceases when channel freezes or is closed".

**Line 35:** Why does lake drainage increase tensile stresses? This should be explained. For "cascading drainage" see my comment above (l 20)

**Line 37**: For which time-period is the number of 26-45% ? And would it be more informative to read that it is ca 35% now but perhaps has increased to this number during a period of xx previous years? I am not sure what to do with this specific, yet not so much saying number of 28-45%.

**Line 38:** specify "these aspects".

**Line 40:** specify "this phenomenon".

**Line 46:** "allow us". I would suggest neutral formulation, as in "large-scale observations allow to "

**Line 48:** "extended time series" -> "extended time periods". Also, was Fitzpatrick et. Al 2013 finally accepted or did It stay in TC Discussion? Pls check journal guidelines if refs to TCD are ok or should be avoided.

**Line 49**: "here, the timing" -> "In these studies, the"

**Line 50:** "and altitude" -> "and lake altitude". Consider mentioning that higher surf temps correlate with lake elevation because surface melting occurs at higher elevations. How is drainage timing affected? Earlier drainage? Later?

**Caption Fig 1**: Add references for Sentinel-2 imagery and ArcticDEM dataset.

**Figure 1 and line 64:** use consistent terminology. Lambert Land/ Lamberts Land?

**Figure 1 and line 68:** You speak of both grounding lines, but 79N's grounding line is not in the map in panel c? And is the grounding line= calving front for Zach? Amend/correct/rephrase

**Line 66:** "This area" Only Zach or also 79N? In that case: "These areas"

**Line 83:** "was calculated" -> "was set"

**Line 84:** "with other researchers" -> "with previous work"

**Line 91:** Are you saying that the lake volume is identical to the volume of water "lost"? Even if the entire lake drains, it is not known whether all water reaches the base or whether some is stored englacially. Care needs to be exerted when formulating methods. Here lake volume calculation is described. How "accurate" is this volume calculation? What is the resolution of the Sentinel pixel, and how sure is one abut the reflectance, and what is the offset 0.5242 and is this specific for Greenland SGLs or fitted to something else? From where comes depth z?

**Line 96:** "ice velocity" -> "ice surface velocity"

**Line 97/101:** "ice sheet velocity" -> "Ice surface velocity"

**Line 100:** Are velocities from July 2019 used for the entire timeperiod (2016-2020)? And only to calculate strain rates? Clarify.

**Line 102 :** "rates of the ice was" -> "rates of the ice, epsilon_1/2, were calculated as (formula)". *So, skip "For this, the first and second invariants of the strain rate tensor was determined in both the x- and y-directions". Also, give equation a number so you can refer to it later in l 186.*

**Line 103:** "rate tensor was" -> "rate tensor were" , x and y before "directions" should be in italics

**Line 105:** " where $\epsilon 1/2$ is the first and second principal strain rates, $\epsilon xx$ is the strain rate tensor in the x-direction, 105 $\epsilon yy$ is the strain rate" -> "where , $\epsilon xx$, $\epsilon yy$ and $\epsilon xy$ are components of the strain rate tensor, and $\epsilon xy$ is know as the thear strain rate."

**Line 116:** "but never rapidly" -> "but did not drain during 2016-2022"

**Line 121:** "found within each" -> "found for each". Also, consider adding a ref to Fig 3 after Fig 2 is mentioned, as eg by writing "shown in Fig. 2, and detailed in Fig 3." This will help address below (l 135)

**Line 122:** I very much like the visualization with the wheel, but find it hard to distinguish the colors in the segments clearly enough to related them to the gradually changing color scale. Consider using a more contrasting scale.

**Line 123-124**: Sentence can be shortened to "Additionally, if the lake never filled up in a certain year, it is marked with an ˋx"

**Line 135:** It is not good practice to skip references to Figures in the main text. Here, Fig. 3f and g are mentioned, but nothing has been said yet about Fig. 3a-e.

**Line 135-236**: I find this sentence somewhat imprecise. Do you mean: Since the lakes are disperses across slow and fast moving areas, some other mechanism than ice deformation (higher in fast flowing regions, so likely faster channel closing) must be responsible for keeping the drainage pathways open so that lakes don't refill? Pls be more specific.

**Line 142:** "temporal variability within each lake" -> "temporal variability of drainage at each lake" ->

**Line 147:** Imprecise formulation. Last week of June cannot be same week near end of July. Reformulate.

**Line 150, 151:** extensive chain drainage is here said to happen on the same day. Why is this not called simultaneous drainage (see line 151) or "cascading" (Christoffersen et al., 2015) or "coupled drainage" (see Otto et al., 2022)? Are chain drainage and simultaneous drainage used as synonyms? Also: "with a couple lakes" -> "with a couple of lakes" (OBS check this throughout the text, there is another occasion where this needs to be fixed in **line 173**)

**Line 152:** Is about Fig 3b. But I see 7 lakes in panel b, but the text is about six lakes? I find panel b quite compact, and it took me a while to match text and what is displayed in panel b. I suggest that lake id's (since you have them anyhow) are added, which would make it easier to refer to the lakes in the describing text (replacing "the lake to the right of the left lakes" etc.) Pls consider adding lake id's and replacing the description in lines 151-155 by clearer, easier to follow text including these lake id's.

**Line 154:** without triggering drainages in them – isn't that to be expected because they are downstream of the other lakes discussed in connection with Fig 3b? Perhaps this info could be added?

**Caption Figure 3d:** "subset, except the lake crossed out, is shown" -> "subset, except the lake crossed out and not included because of its distinctly different drainage pattern, is shown"

**Line 167-169:** Rephrase according to orange, and rephrase non-logic text highlighted in red (an idea cannot be modelled; repetitive use of "idea")

: "drainage after a certain threshold or under certain conditions. An idea that has often been modeled (Alley et al., 2005; Arnold et al., 2014; Banwell et al., 2013; Banwell et al., 2016; Koziol et al., 2017; Tsai and Rice, 2010; van der Veen, 2007) but also recently contradicted by observations (Fitzpatrick et al., 2013; Williamson et al., 2018) is the idea that each lake has" -> "drainage after a certain threshold is passed, or under other certain conditions. An idea that has often been modeled (Alley et al., 2005; Arnold et al., 2014; Banwell et al., 2013; Banwell et al., 2016; Koziol et al., 2017; Tsai and Rice, 2010; van der Veen, 2007) but also recently contradicted by observations (Fitzpatrick et al., 2013; Williamson et al., 2018) is the idea that each lake has"

**Line 170:** Logic again. In a figure, you cannot investigate. But you can show results of what you investigated in a figure. Reformulate.

**Line 171:** Clarification needed: If you say "drain" do you always mean complete drainage? If that is the assumption, drained volume and lake volume are the same. Obs "drained" is not equal to "lost" volume, see an earlier comment.

**Line 173 ff and Fig 4.** I suggest adding a map with the location of these maps. Just by there id's, one has no idea where they are located. Are any of these lakes involved in coupled, chained, or simultaneous drainage? At Ryder Glacier, no coupled lake drainage was observed in 2018 (Otto et al., 2022)

**Line 178:** Use superscripts such that it becomes $km^3$ instead of km3

**Figure 5**: Inconsistent use of units for strain rate, both 1/a and 1/y are used. Check journal requirements re units and correct. Inconsistent use of units continues in **line 209** (and others, check thourghout) where year^-1 is used. Add grounding line for 79N (see comment Fig. 1). Why does legend for strain rate range over a larger interval that shown in panels g and h? High strain regions are not captured currently in panels g and h, unless I am missing something? See also **line 207.**

**Line 189,190:** Edges cannot be strains. Reformulate.

**Line 194-195:** Strain rates can be computed everywhere. What you want to do here is characterize them, not say they exist. Reformulate.

**Line 179:** "found on areas" -> "found in areas"

**Lines 2013-205:** Unnecessary repetition. Rephrase without repetition, and observe that velocity data was acquired for 2019, and that strain data was computed from it. You formulate this correctly in the caption to Fig 5.

**Line 223:** Abbreviation AWS  has not been introduced. Should be done in line 109.

**Figure 6:** Average surface temperature, should that be average air temperature at 2 m above ground? What is the elevation of these AWS? What is an expected elevation corrected temperature over the elevation range of the lakes studied? Perhaps this should be added as an interval around both temperature lines in the figure.

**Caption Fig 6:** Why use "lower" "middle" and "upper" describing categories if you have introduced colors for them? Use "light green", "green" and "blue" instead.

**Line 240-244: Switch: Talk about your results first, then about they confirm what has been reported by others.**

**Figure A3.** Check unit (m/y, m/a, m year^1 etc, see earlier comments. Add grounding lines, see earlier comments. Lamberts vs lambert Land, see earlier comment.

**Line 275:** "lake itself and would thus only drain if" -> lake itself and the lake would thus only drain if"

**Line 276:** why does nearby drainage affect strain rates?

**Line 385**: Shouldn't this be the following?

Lutz, K., Bever, L., Sommer, C., Seehaus, T., Humbert, A., Scheinert, M., and Braun, M.: Assessing supraglacial lake depth using ICESat-2, Sentinel-2, TanDEM-X, and in situ sonar measurements over Northeast and Southwest Greenland, The Cryosphere, 18, 5431–5449, https://doi.org/10.5194/tc-18-5431-2024, 2024.

---

## Referee Comment (RC2)

**Review of "Multi-annual patterns of rapidly draining supraglacial lakes in Northeast Greenland" by Katrina Lutz et al submitted to *The Cryosphere***
Review prepared December 2024

**Overview**
This manuscript investigates the spatial and temporal variability of supraglacial lake drainages on two major glaciers in Northeast Greenland—Zachariae Isstrom and 79N Glacier—during the 2016–2022 melt seasons. Using Sentinel-2 imagery, the study tracks individual lakes to identify drainage events and explores potential correlations with factors such as ice strain rate, elevation, lake volume, and seasonal temperature.

The findings reveal significant variability in drainage patterns, including the occurrence of chain drainages and temporal clustering at higher elevations, but limited correlation with the investigated environmental factors. The authors suggest a critical role of crevasses within lake boundaries as a precondition for rapid drainage events and emphasize the need for higher-resolution remote sensing or in situ data to refine understanding of these mechanisms. This manuscript adds important insights into supraglacial lake dynamics in Greenland, despite lack of finding a coherent mechanism triggering lake drainages.

I found the manuscript to be overall well-designed, referenced, and written, and that it is supported by detailed, yet clear, figures. I think the article is very well-suited for publication in The Cryosphere after a few relatively minor changes. Below I list a couple broader comments that I'd like the authors to consider followed by a few specific comments. I thank the authors in advance for considering these comments.

Sincerely,
Luke Trusel

**Broader comments**
The authors have effectively assessed relationships between lake water volumes and drainage patterns. However, I'd suggest that the study might benefit from incorporating maximum lake depth as the primary parameter, as it more directly relates to hydrofracture potential given that water depth influences the pressure exerted at the lake bottom (e.g., van der Veen, 2007). My concern is that by only looking at lake volume, the analysis could overlook instances where smaller but deep lake lakes possesses a higher propensity for hydrofracture compared to larger but shallower lakes.

My second comment relates to the analysis between summer air temperature and drainage as illustrated in Figure 6. I wonder if the authors may have overlooked a potential explanation for the observed peak in rapid drainages in the warmer (presumably higher-melt) year of 2019, which followed the colder, lower-melt year of 2018. I would expect that the limited meltwater and fewer drainages in 2018 likely resulted in a less efficient subglacial hydrological system due to reduced flushing and connectivity of basal drainage

pathways. In such conditions, a sudden influx of meltwater in the following high-melt year (2019) could have increased basal water pressure and enhanced basal slip in the inefficient basal hydrological system, triggering an ice dynamical response and more drainages. This could also help explain the apparent larger clusters of chain-reaction drainages in 2019 as illustrated in Figure 3. This idea would align with the findings of Stevens et al (2015), where (if I recall correctly) they demonstrate inefficient basal drainage systems after periods of low melt can amplify the effects of subsequent drainage events, including basal slip and tensile stresses that propagate to neighboring lakes.

**Specific comments**
Introduction paragraph 1: While the paragraph overall is well referenced, there are multiple sentences here without supporting references. I'd suggest more specifically connecting the statements in the text to the cited literature rather than just clumping the references together.

L49: Please be more specific to which study "Here" is referencing.

L68: Glacier's -> Glaciers'

L85: Could you please clarify how a rapid drainage was defined. I found the method description here somewhat vague – is it a drainage occurring anywhere from 1 to 10 days, with many being 5 days or less? Some clarity in the description would be helpful. You may also consider stating (or mapping) the average time constraint between lake observations related to drainages.

L102+L103: Was -> were

L121: Some words out of place here: change to "The temporal and spatial variations" (or similar)

L148: Add "near-" before simultaneously?

Figure 3: Just commenting to say this is a very nice figure with interesting results!

L178: Make 3's superscript.

L203-206: The two sentences here repeat. Delete one.

L222 (and elsewhere): These are presumably summer surface **air** temperatures, correct? Surface temperature alone implies the skin temperature rather than near-surface air temperature.

L278-281: The couple sentences following "Upon…" are confusing to me. The first implies that the lakes are too small for a fracture to stay within the lake for more than one year,

whereas the second says the lakes are large enough to have the fracture stay within the lake. Could you please clarify these statements?

**References from this review**

Stevens, L. A., Behn, M. D., McGuire, J. J., Das, S. B., Joughin, I., Herring, T., Shean, D. E., and King, M. A.: Greenland supraglacial lake drainages triggered by hydrologically induced basal slip, Nature, 522, 73–76, https://doi.org/10.1038/nature14480, 2015.

van der Veen, C. J.: Fracture propagation as means of rapidly transferring surface meltwater to the base of glaciers, Geophys. Res. Lett., 34, L01501, https://doi.org/10.1029/2006GL028385, 2007.

---

## Author Response (AR1)

**Point-by-point response to comments provided by Luke Trusel for "Multi-annual patterns of rapidly draining supraglacial lakes in Northeast Greenland"**

**Broader comments**

*The authors have effectively assessed relationships between lake water volumes and drainage patterns. However, I'd suggest that the study might benefit from incorporating maximum lake depth as the primary parameter, as it more directly relates to hydrofracture potential given that water depth influences the pressure exerted at the lake bottom (e.g., van der Veen, 2007). My concern is that by only looking at lake volume, the analysis could overlook instances where smaller but deep lake lakes possesses a higher propensity for hydrofracture compared to larger but shallower lakes.*

The idea to look at the maximum lake depth as a controlling factor for lake drainage is indeed a valid point due to the reasons you mention. Thus, we calculated the max depth before drainage or the largest max depth achieved throughout a melt season in which the lake did not drain, the results of which can be seen in the figure below. In general, it appears that there is little variation in the max depths achieved throughout the melt seasons, whereas the difference in volumes is clearly apparent (Fig. 4). The reasons for this are most likely due to the depth estimation equation we use, which was only developed on data up to 8 m, and which reaches a reflectance saturation point around 8.5 m. Since a lake's maximum depth is likely to be as deep or deeper than that, many of these max depth values are similar. This limitation does not have as strong of an influence on the total volume, as this depth saturation only affects the deepest areas of a lake. Furthermore, it is not necessarily the case that the max depth will occur near the region of hydrofracture. This can be seen through partial drainages, where the deepest part of the lake does not drain and thus the max depth remains relatively constant despite a rapid drainage having occurred. This was seen for lake 381 in a few melt seasons in particular. Also, looking at the max depths of the lakes in this figure, there were several instances where the max depth was reached several weeks before a drainage occurred, implying that the pressure from the deepest areas alone are not triggering the drainage. Overall, since these specific lakes do not form in large, shallow formations, we would presume that using the volume as an indicator of load on the lake bed is justified based on these findings.

[Figure]

*My second comment relates to the analysis between summer air temperature and drainage as illustrated in Figure 6. I wonder if the authors may have overlooked a potential explanation for the observed peak in rapid drainages in the warmer (presumably highermelt) year of 2019, which followed the colder, lower-melt year of 2018. I would expect that the limited meltwater and fewer drainages in 2018 likely resulted in a less efficient subglacial hydrological system due to reduced flushing and connectivity of basal drainage pathways. In such conditions, a sudden influx of meltwater in the following high-melt year (2019) could have increased basal water pressure and enhanced basal slip in the inefficient basal hydrological system, triggering an ice dynamical response and more drainages. This could also help explain the apparent larger clusters of chain-reaction drainages in 2019 as illustrated in Figure 3. This idea would align with the findings of Stevens et al (2015), where (if I recall correctly) they demonstrate inefficient basal drainage systems after periods of low melt can amplify the effects of subsequent drainage events, including basal slip and tensile stresses that propagate to neighboring lakes.*

Thank you for sharing this insight with us. We had indeed discussed the influence of local stressors on the drainages but had not thought of the inefficient drainages systems following a cold year to be a factor behind it, especially regarding the high drainage rate in 2019. We have now added several sentences in the discussion detailing this idea as a potential influencer.

**Specific comments**

*Introduction paragraph 1: While the paragraph overall is well referenced, there are multiple sentences here without supporting references. I'd suggest more specifically connecting the statements in the text to the cited literature rather than just clumping the references together.*

Several sentences have now been added to the introduction to provide more detail and contextualization. Additionally, more references have been added to support individual statements instead of having large clumps of references. One larger clump of references remains, since breaking these references into individual sentences would substantially expand the paragraph and deter from the flow and focus of the paragraph.

*L49: Please be more specific to which study "Here" is referencing.*
Clarified.

*L68: Glacier's -> Glaciers'*
Corrected.

*L85: Could you please clarify how a rapid drainage was defined. I found the method description here somewhat vague – is it a drainage occurring anywhere from 1 to 10 days, with many being 5 days or less? Some clarity in the description would be helpful. You may also consider stating (or mapping) the average time constraint between lake observations related to drainages.*
Yes, we regard a rapid drainage in theory to be occurring in less than two days, but because remote sensing methods often do not provide clear imagery within that timeframe, the 'acceptable' interval was extended to be 10 days. I have calculated the average image interval (2.8 days) and adjusted the text to clarify that.

*L102+L103: Was -> were*
Corrected.

*L121: Some words out of place here: change to "The temporal and spatial variations" (or similar)*
Corrected.

*L148: Add "near-" before simultaneously?*
Corrected.

*L178: Make 3's superscript.*
Corrected.

*L203-206: The two sentences here repeat. Delete one.*
Corrected.

*L222 (and elsewhere): These are presumably summer surface air temperatures, correct? Surface temperature alone implies the skin temperature rather than near-surface air temperature.*
Corrected in the text and the figure.

*L278-281: The couple sentences following "Upon…" are confusing to me. The first implies that the lakes are too small for a fracture to stay within the lake for more than one year, whereas the second says the lakes are large enough to have the fracture stay within the lake. Could you please clarify these statements?*
I can understand the confusion. The first sentence was referring to lakes in high strain regions, whereas the second sentence was referring to frequently draining lakes, which are often not in high strain regions in our analysis. I have adjusted the text to hopefully make that contrast more clear.

**Point-by-point response to comments provided by Nina Kirchner for "Multi-annual patterns of rapidly draining supraglacial lakes in Northeast Greenland"**

**Assessment and specific comments**

*To start with, a stronger motivation should be given why this dataset was created and analysed. Formulations such as "rapid drainages can substantially affect [...] aspects of the glacial environment throughout the melt season" are too much of a commonplace and don't match the level expected from a manuscript submitted to TC.*
Several sentences have now been added to strengthen the motivation of this research and give more concrete details about previous work. Additionally, more references have been added to substantiate these statements. Finally, several sentences have been reformulated to achieve a higher and more eloquent register.

*I recommend that a comment on terminology regarding the simultaneous (raid) drainage of supraglacial lakes is included. To my knowledge, these events have first been referred to as "cascading drainage" (Christoffersen et al., 2015, cited in the manuscript), then "coupled drainage (Otto et al., 2022. Supraglacial lake expansion, intensified lake drainage frequency, and first observation of coupled lake drainage, during 1985-2020 at Ryder Glacier, N Greenland. Front Earth Sci, doi.org/10.3389/feart.2022.978137), and then here as "chain drainages". It would be beneficial in my view to adopt an already existing terminology (unless "chained" exists already, but in that case I miss a reference to it).*
To enhance cohesion with previous studies, we have used the more consistent terminology of "cascading drainage" throughout the manuscript.

*Continuing on this, it needs to be pointed out that the work of Otto et al. (2022), doing a similar study at Ryder Glacier, has apparently been overlooked. As Otto's work covers the time-span 1985-2020, including first observed coupled drainage at Ryder Glacier, I had expected to read about a comparison of drainage of SGLs at northern Greenland glaciers at least in the Discussion section?*
This manuscript was indeed overlooked. Due to the relevance of the analysis presented in Otto et al. (2022), this manuscript has been added to the discussion section in relation to coupled drainage occurrence and rapid drainage expansion.

*Likewise, I am curious how rapid drainage events contrast to slow drainage events. I assume that to identify "rapid" drainage, all types of drainage need to be detected first? I realize that including a detailed comparison of slow vs rapid drainage is likely beyond the scope of the manuscript, but it would be helpful to know e.g how large the fraction of rapidly draining lakes is compared to slowly draining (or mentioning other similar simple comparative metrics). For southwest Greenland, such info is cited from the literature, but how is it for the region studied here? I think info is available (from e.g Fig 1 which indicates non-rapidly draining lakes), so could be included in a straightforward manner, and even be given some thought and reflection in the discussion section.*
We can understand the interest in such a comparison; however, through our data setup, it would be quite cumbersome and inaccurate to quantify slow drainages events. The volume of each lake was tracked over the melt season, which in theory would allow slow drainages to be identified. However, we found it difficult to precisely quantify the exact conditions for which a slow drainage could be identified, since the entire lake often does not fully drain when drained slowly and the volumes of lakes can fluctuate throughout the melt season, making it difficult to discern slow drainages from temperature-driven fluctuation. Thus, we will exclude the analysis of slow drainages from this study.

*Finally, I think it is assumed in the manuscript (unless I misunderstood) that all drained water reaches the bed. But what about englacial storage, that is, if the drained volume vanishes from the surface but if not all of it reaches the bed? Could this impact subsequent drainage in one way or the other? If crevasses in the lake would aid rapid drainage, maybe englacial storage would hinder it? Maybe englacial storage could be reflected on in the discussion, I would be curious to hear the author's thoughts on it.*
While it would indeed be interesting to be able to provide insight into the routing and possible storage of meltwater englacially, such an analysis is out of the scope of a remote sensing-based manuscript. Postulations about the potential influence of englacial storage on rapid drainages would not be appropriate, as we do not have scientific evidence on which we could base any claims.

**Detailed comments and questions**
*Line 9-10: "Supraglacial lakes are known to undergo rapid drainages in which the contents of the lake are drained through ice hydrofracture to the glacier bed" ... -> "Supraglacial lakes are known to undergo rapid drainages in which their watermasses are drained through ice hydrofracture to the glacier bed..."*
Adjusted.

*Line 10-11: "Despite the impact of this sudden loss of meltwater from the glacier, the conditions.. .." -> "Despite the impact of this sudden englacial transport of meltwater, the conditions…"*
Adjusted.

*Line 13: "Over the 2016 - 2022 summer melt seasons, each supraglacial lake was tracked "--- ->*
*"Over the 2016 - 2022 summer melt seasons, supraglacial lakes on these glaciers were tracked" Question: Over the entire glaciers or with focus on specific regions? This info comes later (l 63) but should be given here.*
Adjusted.

*Lines 17-20, from "Furthermore": these sentences are unclear/vague and need to be reformulated*
Reformulated.

*Line 20: physical location -> spatial position ?*
Adjusted.

*Line 20: Chain drainage. I think you may be referring to "cascading" or "coupled drainage", as earlier described by Christoffersen et al., 205, and Otto et al., 2022. If yes, perhaps coupled drainage should be used here, too as terminology. If however chain drainage differs from coupled drainage, it should be explained (later) how.*
This has been changed to "cascading drainage" in order to be consistent with previous studies.

*Line 20: unclear wording in "in which more than one neighboring lake" – neighboring to what? You mena to another lake. You could rewrite as "in which neighboring lakes"*
Adjusted.

*Line 23: "would need to be filled to allow for a rapid drainage to occur, particularly the existence of a crevasse within" -> "would need to be met to allow for a rapid drainage to occur, particularly the existence of fractures or crevasses within". In any case, I suggest that something in plural form is needed. Not just one crevasse.*
Adjusted.

*Line 27: References are needed re better understood interconnectedness of supra- and subglacial hydrology*
A few references have been added to support this statement.

*Line 29: "where the lakes" -> where a lake"*
Adjusted.

*Line 30: "routed vertically" -> "routed towards the bed". Note that crevasses may not extend all the way to the bed, and that meltwater may be stored englacial. Note also that neither moulins nor crevasses are vertical.*
Adjusted.

*Line 31: "Conduits remain open… until the channel freezes or is closed". I would rephrase this differently, like "meltwater supplies to bed ceases when channel freezes or is closed".*
Adjusted.

*Line 35: Why does lake drainage increase tensile stresses? This should be explained. For "cascading drainage" see my comment above (l 20)*
An explanation has been added.

*Line 37: For which time-period is the number of 26-45% ? And would it be more informative to read that it is ca 35% now but perhaps has increased to this number during a period of xx previous years?*
*I am not sure what to do with this specific, yet not so much saying number of 28-45%.*
These values encompass the span of values found among different manuscripts; however, that was indeed not made clear here. I have rephrased it to be more of an approximation and included the other citations contributing to the range of values.

*Line 38: specify "these aspects".*
This sentence has been changed to be more detailed.

*Line 40: specify "this phenomenon".*
Adjusted.

*Line 46: "allow us". I would suggest neutral formulation, as in "large-scale observations allow to "*

Adjusted.

*Line 48: "extended time series" -> "extended time periods". Also, was Fitzpatrick et. Al 2013 finally accepted or did It stay in TC Discussion? Pls check journal guidelines if refs to TCD are ok or should be avoided.*
Adjusted. I do not see this article anywhere besides The Cryosphere Discussion. According to The Cryosphere guidelines, preprints with a DOI are acceptable to reference.

*Line 49: "here, the timing" -> "In these studies, the"*
Adjusted.

*Line 50: "and altitude" -> "and lake altitude". Consider mentioning that higher surf temps correlate with lake elevation because surface melting occurs at higher elevations. How is drainage timing affected? Earlier drainage? Later?*
This information has now been integrated into the text.

*Caption Fig 1: Add references for Sentinel-2 imagery and ArcticDEM dataset.*
The reference for the ArcticDEM dataset has been added. We acknowledge the use of ESA's Sentinel-2 imagery in the acknowledgements.

*Figure 1 and line 64: use consistent terminology. Lambert Land/ Lamberts Land?*
The terminology in the figure has been adjusted.

*Figure 1 and line 68: You speak of both grounding lines, but 79N's grounding line is not in the map in panel c? And is the grounding line= calving front for Zach? Amend/correct/rephrase*
The text and figure have been adjusted.

*Line 66: "This area" Only Zach or also 79N? In that case: "These areas"*
Adjusted.

*Line 83: "was calculated" -> "was set"*
Adjusted.

*Line 84: "with other researchers" -> "with previous work"*
Adjusted.

*Line 91: Are you saying that the lake volume is identical to the volume of water "lost"? Even if the entire lake drains, it is not known whether all water reaches the base or whether some is stored englacially. Care needs to be exerted when formulating methods. Here lake volume calculation is described. How "accurate" is this volume calculation? What is the resolution of the Sentinel pixel, and how sure is one abut the reflectance, and what is the offset 0.5242 and is this specific for Greenland SGLs or fitted to something else? From where comes depth z?*
This was indeed not clearly stated – the pre- and post-drainage volumes were calculated and differenced to estimate how much water was lost from the lake. This is now described in the text, along with the note that the water can be either routed to the glacier bed or be stored englacially. More details of this depth equation can be found in Lutz et al. (2024). This equation was optimally fit to sonar-based depth data gathered in Northeast Greenland. The offset is most likely due to the large variations is surface reflection in very shallow water due to the influence of sediment on the glacier surface.

*Line 96: "ice velocity" -> "ice surface velocity"*
Adjusted.

*Line 97/101: "ice sheet velocity" -> "Ice surface velocity"*
Adjusted.

*Line 100: Are velocities from July 2019 used for the entire timeperiod (2016-2020)? And only to calculate strain rates? Clarify.*
Clarified.

*Line 102 : "rates of the ice was" -> "rates of the ice, epsilon_1/2, were calculated as (formula)". So, skip "For this, the first and second invariants of the strain rate tensor was determined in both the x and y-directions". Also, give equation a number so you can refer to it later in l 186.*
Adjusted. The equation number was added in line 186.

*Line 103: "rate tensor was" -> "rate tensor were" , x and y before "directions" should be in italics*
Adjusted.

*Line 105: " where $\epsilon 1/2$ is the first and second principal strain rates, $\epsilon xx$ is the strain rate tensor in the x-direction, 105 $\epsilon yy$ is the strain rate" -> "where , $\epsilon xx$, $\epsilon yy$ and $\epsilon xy$ are components of the strain rate tensor, and $\epsilon xy$ is know as the thear strain rate."*
Adjusted.

*Line 116: "but never rapidly" -> "but did not drain during 2016-2022"*
Adjusted.

*Line 121: "found within each" -> "found for each". Also, consider adding a ref to Fig 3 after Fig 2 is mentioned, as eg by writing "shown in Fig. 2, and detailed in Fig 3." This will help address below (l 135)*
Adjusted.

*Line 122: I very much like the visualization with the wheel, but find it hard to distinguish the colors in the segments clearly enough to related them to the gradually changing color scale. Consider using a more contrasting scale.*
We understand the neighboring colors are sometimes difficult to distinguish, but with so many time divisions, it is difficult to find a color scale that linearly changes while being distinct enough between divisions. As this map is visually meant to be used more qualitatively, we believe this scale is sufficient.

*Line 123-124: Sentence can be shortened to "Additionally, if the lake never filled up in a certain year, it is marked with an 'x"*
We think the rest of the sentence provides clarifying information.

*Line 135: It is not good practice to skip references to Figures in the main text. Here, Fig. 3f and g are mentioned, but nothing has been said yet about Fig. 3a-e.*
Text was adjusted according to previous comment (line 121).

*Line 135-236: I find this sentence somewhat imprecise. Do you mean: Since the lakes are disperses across slow and fast moving areas, some other mechanism than ice deformation (higher in fast flowing regions, so likely faster channel closing) must be responsible for keeping the drainage pathways open so that lakes don't refill? Pls be more specific.*
Yes, that is what we meant. The text has been adjusted to make that more clear.

*Line 142: "temporal variability within each lake" -> "temporal variability of drainage at each lake" ->*
Adjusted.

*Line 147: Imprecise formulation. Last week of June cannot be same week near end of July. Reformulate.*
The weeks were not meant to be the same; they were both meant to demonstrate that the lakes within each group drained within the same week as the other lakes in the same group. In other words, all lakes in Fig. 3d tend to drain within a specific week and independently the lakes in Fig. 3e tend to drain within another week each year. This has been reformulated to make this distinction more clear.

*Line 150, 151: extensive chain drainage is here said to happen on the same day. Why is this not called simultaneous drainage (see line 151) or "cascading" (Christoffersen et al., 2015) or "coupled drainage" (see Otto et al., 2022)? Are chain drainage and simultaneous drainage used as synonyms? Also: "with a couple lakes" -> "with a couple of lakes" (OBS check this throughout the text, there is another occasion where this needs to be fixed in line 173)*
The terminology of "chain drainage" has been replaced with "cascading drainage" to enhance text clarity. Additionally, the noted typos have been corrected.

*Line 152: Is about Fig 3b. But I see 7 lakes in panel b, but the text is about six lakes? I find panel b quite compact, and it took me a while to match text and what is displayed in panel b. I suggest that lake id's (since you have them anyhow) are added, which would make it easier to refer to the lakes in the describing text (replacing "the lake to the right of the left lakes" etc.) Pls consider adding lake id's and replacing the description in lines 151-155 by clearer, easier to follow text including these lake id's.*
Lake IDs have been added to panel b in Figure 3 and the text has been adjusted accordingly.

*Line 154: without triggering drainages in them – isn't that to be expected because they are downstream of the other lakes discussed in connection with Fig 3b? Perhaps this info could be added?*

It is indeed more likely that cascading drainages would be triggered downstream. This presumption has been added to the text.

*Caption Figure 3d: "subset, except the lake crossed out, is shown" -> "subset, except the lake crossed out and not included because of its distinctly different drainage pattern, is shown"*
Adjusted.

*Line 167-169: Rephrase according to orange, and rephrase non-logic text highlighted in red (an idea cannot be modelled; repetitive use of "idea"): "drainage after a certain threshold or under certain conditions. An idea that has often been modeled (Alley et al., 2005; Arnold et al., 2014; Banwell et al., 2013; Banwell et al., 2016; Koziol et al., 2017; Tsai and Rice, 2010; van der Veen, 2007) but also recently contradicted by observations (Fitzpatrick et al., 2013; Williamson et al., 2018) is the idea that each lake has" -> "drainage after a certain threshold is passed, or under other certain conditions. An idea that has often been modeled (Alley et al., 2005; Arnold et al., 2014; Banwell et al., 2013; Banwell et al., 2016; Koziol et al., 2017; Tsai and Rice, 2010; van der Veen, 2007) but also recently contradicted by observations (Fitzpatrick et al., 2013; Williamson et al., 2018) is the idea that each lake has"*
Adjusted.

*Line 170: Logic again. In a figure, you cannot investigate. But you can show results of what you investigated in a figure. Reformulate.*
Adjusted.

*Line 171: Clarification needed: If you say "drain" do you always mean complete drainage? If that is the assumption, drained volume and lake volume are the same. Obs "drained" is not equal to "lost" volume, see an earlier comment.*
When the phrase "drained" is used, it refers to meltwater leaving the surface through drainage. In this instance, however, it is more sensible to evaluate the total volume before drainage when determining if there is a volume threshold above which the pressure induces hydrofracture. Thus, the phrasing in the text and the values in the figure have been adjusted to reflect that. The values are generally only minimally different, since the majority of drainages fully emptied or nearly fully emptied the lake basin.

*Line 173 ff and Fig 4. I suggest adding a map with the location of these maps. Just by there id's, one has no idea where they are located. Are any of these lakes involved in coupled, chained, or simultaneous drainage? At Ryder Glacier, no coupled lake drainage was observed in 2018 (Otto et al., 2022)*
The IDs of these lakes have been added to the overview map in Fig. 2 for clarity. All but one of these lakes was involved in a cascading drainage. One of these lakes was involved in a cascading drainage with three other lakes in 2018.

*Line 178: Use superscripts such that it becomes km$^3$ instead of km3*
Adjusted.

*Figure 5: Inconsistent use of units for strain rate, both 1/a and 1/y are used. Check journal requirements re units and correct. Inconsistent use of units continues in line 209 (and others, check thourghout) where year^-1 is used. Add grounding line for 79N (see comment Fig. 1). Why does legend for strain rate range over a larger interval that shown in panels g and h? High strain regions are not captured currently in panels g and h, unless I am missing something? See also line 207.*
The units have been changed to be consistent as 1/a. The strain rate legend is so much larger than the values seen in panels g and h because the highest points of strain on the maps are not located directly under any lakes. The area where the maximum strain rate values are found are minimal. Line 207 has been slightly rephrased to make this clearer.

*Line 189,190: Edges cannot be strains. Reformulate.*
The phrase "edge" was referring to the location on the glacier, not the strain itself. This sentence has been reformulated to make that more clear.

*Line 194-195: Strain rates can be computed everywhere. What you want to do here is characterize them, not say they exist. Reformulate.*
Rephrased.

*Line 179: "found on areas" -> "found in areas"*
Adjusted.

*Lines 2013-205: Unnecessary repetition. Rephrase without repetition, and observe that velocity data was acquired for 2019, and that strain data was computed from it. You formulate this correctly in the caption to Fig 5.*
Adjusted.

*Line 223: Abbreviation AWS has not been introduced. Should be done in line 109.*
Adjusted.

*Figure 6: Average surface temperature, should that be average air temperature at 2 m above ground? What is the elevation of these AWS? What is an expected elevation corrected temperature over the elevation range of the lakes studied? Perhaps this should be added as an interval around both temperature lines in the figure.*
Yes, these are average surface air temperatures - this phrasing has been corrected. Additionally, the elevations of the AWSs have been added to the text along with an estimated elevation-corrected temperature range covering the elevations where lakes are present through the calculation of local time lapse rates for each melt season. This range is now displayed in the figure.

*Caption Fig 6: Why use "lower" "middle" and "upper" describing categories if you have introduced colors for them? Use "light green", "green" and "blue" instead.*
 The terms "lower", "middle", and "upper" were used to accommodate readers with colorblindness, but we have added the colors in parentheses to make it more clear for those unaffected with colorblindness.

*Line 240-244: Switch: Talk about your results first, then about they confirm what has been reported by others.*
Thank you for the suggestion, but we do not think restructuring these sentences would improve the readability or logic flow.

*Figure A3. Check unit (m/y, m/a, m year^1 etc, see earlier comments. Add grounding lines, see earlier comments. Lamberts vs lambert Land, see earlier comment.*
The figure has been updated.

*Line 275: "lake itself and would thus only drain if" -> lake itself and the lake would thus only drain if"*
Adjusted.

*Line 276: why does nearby drainage affect strain rates?*
This was meant to say tensile stress; it has now been corrected.

*Line 385: Shouldn't this be the following?*
*Lutz, K., Bever, L., Sommer, C., Seehaus, T., Humbert, A., Scheinert, M., and Braun, M.: Assessing supraglacial lake depth using ICESat-2, Sentinel-2, TanDEM-X, and in situ sonar measurements over Northeast and Southwest Greenland, The Cryosphere, 18, 5431–5449, https://doi.org/10.5194/tc-18-5431-2024, 2024.*
Yes, the manuscript had not yet finished the review process when this manuscript had been submitted. The citation has been updated.